# Transcriptomics and Metabolomics of Reactive Oxygen Species Modulation in Near-Null Magnetic Field-Induced *Arabidopsis thaliana*

**DOI:** 10.3390/biom12121824

**Published:** 2022-12-06

**Authors:** Ambra S. Parmagnani, Giuseppe Mannino, Massimo E. Maffei

**Affiliations:** Department of Life Sciences and Systems Biology, University of Turin, Via G. Quarello 15/a, 10135 Turin, Italy

**Keywords:** reduction of geomagnetic field, gene expression, metabolomics, flavonoids, isoflavonoids, hydrogen peroxide

## Abstract

The geomagnetic field (GMF) is a natural component of Earth’s biosphere. GMF reduction to near-null values (NNMF) induces gene expression modulation that generates biomolecular, morphological, and developmental changes. Here, we evaluate the effect of NNMF on gene expression and reactive oxygen species (ROS) production in time-course experiments on *Arabidopsis thaliana*. Plants exposed to NNMF in a triaxial Helmholtz coils system were sampled from 10 min to 96 h to evaluate differentially expressed genes (DEGs) of oxidative stress responses by gene microarray. In 24–96 h developing stages, H_2_O_2_ and polyphenols were also analyzed from roots and shoots. A total of 194 DEGs involved in oxidative reactions were selected, many of which showed a fold change ≥±2 in at least one timing point. Heatmap clustering showed DEGs both between roots/shoots and among the different time points. NNMF induced a lower H_2_O_2_ than GMF, in agreement with the expression of ROS-related genes. Forty-four polyphenols were identified, the content of which progressively decreased during NNMF exposition time. The comparison between polyphenols content and DEGs showed overlapping patterns. These results indicate that GMF reduction induces metabolomic and transcriptomic modulation of ROS-scavenging enzymes and H_2_O_2_ production in *A. thaliana*, which is paralleled by the regulation of antioxidant polyphenols.

## 1. Introduction

A growing body of evidence indicates that plants react to varying magnetic field (MF) fluxes at values both below and above the geomagnetic field (GMF) [1,2,3]. Reduction of the GMF (20–60 µT) to Near Null Magnetic Field (NNMF, about 30 nT) has been shown to influence many plant biological processes [4,5].

In living systems, four different mechanisms of magnetoperception have been described: (i) the radical pairs mechanism (i.e., magnetically sensitive chemical intermediates that are formed by photoexcitation of cryptochrome [6,7]), which is present in animals [8], humans [9] and plants [10]; (ii) the presence of MF sensory receptors described in magnetotactic bacteria [11]; (iii) the presence of electroreceptors in elasmobranch animals [12] and; (iv) the biocompass model based on MagR/Cry complex [13]. Among the four possible mechanisms of magnetoreception, at least two (the radical pair mechanism (RPM) of chemical magnetosensing and the MagR/Cry biocompass) adequately explain the alterations in the MF by the rates of redox reactions and subsequently altered concentrations of free radicals and reactive oxygen species (ROS) observed in plants [10,14,15,16,17,18].

Cryptochrome (Cry) modulates ROS in response to weak MFs through altering the rate of redox reactions in the presence of a MF [10,19]. This phenomenon affects the cellular ROS production (in the nucleus and cytosol, and possibly also in other organelles as well) and is proposed to be similar in both plants and animals [3,18]. This mechanism perfectly predicts an effect on cellular ROS signaling pathways; therefore, this hypothesis explains the ROS-related modulation of gene expression in response to MFs that has been observed in recent studies [20]. MF type/intensity/frequency, exposure time and assay time point, as well as different biological samples have been demonstrated to induce different ROS levels [21].

It is assumed that the singlet recombination of the FADH^•^/O_2_^•−^-RPM produces hydrogen peroxide (H_2_O_2_), possibly via the C4a-hydroperoxy-flavin, and that C^•^ re-oxidizes FADH^•^ to FAD [22]. It has also been proposed that that O_2_^•−^ and H_2_O_2_ production in some metabolic processes occur through singlet–triplet modulation of semiquinone flavin (FADH^•^) enzymes and O_2_^•−^ spin-correlated radical pairs [23].

Therefore, the aim of this work was to explore the transcriptomics and metabolomics of ROS production/metabolism by time-course gene expression analysis in both roots and shoots and to evaluate the production of both ROS and scavenging biomolecules in *Arabidopsis thaliana* plants exposed to NNMF conditions.

## 2. Materials and Methods

### 2.1. Plant Materials and Growth Conditions

*Arabidopsis thaliana* ecotype Columbia-0 (Col-0) wild-type (WT) seeds were surface-sterilized with 70% (*v*/*v*) ethanol for 15 min and then quickly washed with 100% (*v*/*v*) ethanol 2 times. Seeds were individually plated in sterile conditions on half-strength Murashige and Skoog (MS) medium [24] supplemented with 0.1% (*w*/*v*) sucrose, 0.05% (*w*/*v*) MES, pH 5.8, and 0.8% (*w*/*v*) plant agar. Plates were sealed with Micropore tape to allow gas exchange and avoid condensation. Plates were then vernalized and exposed to light for 14 h before being kept in darkness as previously described [20]. Plates were then transferred under either NNMF or GMF and exposed to 120 μmol m^−2^ s^−1^ light provided by a tunable LED lighting system source (PHYTOFY RL 150W, Osram, München, Germany) at 22 °C (±1.5 °C) with a 16/8 light/darkness photoperiod. For the entire experimental time, the temperature was set and maintained by air-conditioning. All experiments were performed under normal gravity and atmospheric pressure.

### 2.2. Near Null Magnetic Field (NNMF) Generation System and Plant Exposure

The GMF (or local geomagnetic field) values where typical of the northern hemisphere at 45°0′59″ N and 7°36′58″ E coordinates. Near-null magnetic field (NNMF) was generated as previously described [25]. Real-time monitoring of the MF in the plant exposure chamber was achieved with a three-axis magnetic field sensor (model Mag-03, Bartington Instruments, Oxford, UK) that was placed at the geometric center of the Helmholtz coils. The output data from the magnetometer were uploaded to a VEE Pro for Windows software release 7.51 (Agilent Technologies, Santa Clara, CA, USA; https://www.keysight.com/it accessed on 1 October 2022) to accurately adjust the current applied through each of the Helmholtz coil pairs in order to maintain the MF constant inside the plant growth chamber at NNMF intensity, as recently reported [20]. Plates containing *A. thaliana* seedlings seeded 134 h before were placed in the geometric center of the triaxial Helmholtz coils system and exposed either to NNMF for 10 min, 1 h, 2 h, 4 h, 24 h, 48 h, and 96 h. After the exposure period, shoots and roots were separately harvested and immediately frozen in liquid nitrogen.

### 2.3. RNA Extraction from Arabidopsis Shoots and Roots upon Time-Course Exposure to GMF and NNMF

For each time point, 100 mg of frozen *A. thaliana* roots or shoots exposed to either GMF or NNMF were ground in liquid nitrogen and total RNA was isolated and its quality and quantity checked as previously described [20]. Quantification of RNA was also confirmed spectrophotometrically by using a NanoDrop ND-1000 (Thermo Fisher Scientific, Waltham, MA, USA).

### 2.4. cDNA Synthesis and Gene Microarray Analyses (Including MIAME)

Five hundred nanograms of total RNA from each sample were separately reverse-transcribed into double-stranded cDNA, which was transcribed into antisense cRNA and labelled with either Cy3-CTP or Cy5-CTP fluorescent dyes for 2 h at 40 °C as reported before [21]. Cyanine-labeled cRNAs were purified using RNeasy Minikit (Qiagen, Hilden, Germany). Purity and dye incorporation were assessed as previously described [20]. Then, 825 ng of control Cy3-RNAs and 825 ng of treated Cy5-RNAs were pooled together and hybridized using the Gene Expression Hybridization Kit (Agilent Technologies) onto 4 × 44 K *A. thaliana* (v3) Oligo Microarray (Agilent Technologies). The microarray experiment followed a direct 2 × 2 factorial two-color design. This resulted in two-color arrays, satisfying Minimum Information About a Microarray Experiment (MIAME) requirements [26].

Microarrays were scanned as reported before [20] (Agilent Technologies).

GO enrichment information for the differently expressed probe sets was obtained from The *A. thaliana* information resource (https://www.arabidopsis.org/index.jsp accessed on 1 October 2022).

### 2.5. Validation of Microarray Data by Real-Time PCR

In order to validate microarray data, qPCR analyses were performed on a QuantStudio 3 Real-Time PCR System (Applied Biosystems, Foster City, CA, USA) using the same cDNA products obtained as previously described (Section 2.4). Amplification was carried out with the primers listed in Appendix A, while the reaction was performed using Maxima SYBR Green/ROX qPCR Master Mix Kit (Thermo Fisher Scientific, Waltham, MA, USA) in 25 µL of total volume. The thermocycling was performed using two-step cycling protocol: 50 °C for 2 min, 95 °C for 10 min, 95 °C for 15 s, and 60 °C for 60 s. The lasts two step were repeated for 40 cycles. All primers were designed using Primer 3 software. Four different reference genes (cytoplasmic glyceraldehyde-3-phosphate dehydrogenase, (*GAPC2*, *At1g13440*), ubiquitin specific protease 6 (*UBP6*, *At1g51710*), actin1 (*ACT1*, *At2g37620*), and the elongation factor 1B alpha-subunit 2 (*eEF1Balpha2*, *At5g19510*)) were used to normalize the results of qPCR; the most stable gene was the ubiquitin specific protease 6 (*UBP6*). All amplification plots were analyzed with QuantStudio Design & Analysis software (Applied Biosystem, Foster City, CA, USA) to obtain Ct values. Relative RNA levels were calibrated and normalized with the level of the ubiquitin specific protease 6 (*UBP6*) mRNA.

### 2.6. Reactive Oxygen Species Quantification and Activity

Determination of the H_2_O_2_ content was carried out by using MAK311 Peroxide Assay Kit (Sigma-Aldrich, St. Louis, MI, USA). Roots and shoots of *A. thaliana* plants were sampled at different time points as described in paragraphs 2.1 and 2.2. Samples were grinded and extracted in 1:10 (*w*/*v*) mQ water. After centrifugation at 15,000× *g* for 10 min, the supernatant was used for the assay. Four µL of each standard and sample were incubated for 30 min at RT with 20 µL of assay kit Detection Reagent. An optical density of 2 µL was measured at 585 nm with BioSpec-nano Spectrophotometer (Shimadzu, Kyoto, Japan).

### 2.7. Analysis of Antioxidant Molecules by HPLC-ESI-MS/MS

The same water-extracted samples used for ROS quantification (see Section 2.6) were immediately injected into an HPLC system (Agilent Technologies 1200, Santa Clara, CA, USA) coupled to a diode array detector (DAD) (Agilent Technologies 1200, Santa Clara, CA, USA). Additionally, analyses were performed by using a mass spectrometer (6330 Series Ion Trap Mass Spectrometer System, Agilent Technologies, Santa Clara, CA, USA) coupled to an electrospray ionization (ESI) module. The chromatographic separation was carried out using a constant flow rate (0.2 mL min^−1^) through a reverse phase C18 Luna column (3.00 μm, 150 × 3.0 mm i.d., Phenomenex, CA, USA), maintained at 25 °C by an Agilent 1100 HPLC G1316A Column Compartment. The chromatographic separation was obtained by a binary solvent system consisting of MilliQ H_2_O acidified with 0.1% formic acid (*v*/*v*) (Solvent A) and Acetonitrile acidified with 0.1% formic acid (*v*/*v*) (Sigma-Aldrich, CA, USA) (Solvent B). The initial concentration of the solvents was set at 90% A and 10% B for 5 min, then the concentration of Solvent B was raised to 55% A in 25 min, and finally at 70% B in 25 min. The initial solvent concentration was restored at the end of each run and maintained for an additional 10 min before the next injection. Sample injection volume was set at 5 μL. For each injection, UV–VIS spectra were recorded at 220, 280, 360, and 520 nm, along with the whole UV/Vis spectrum between 200 and 800 nm. Tandem mass spectrometry analyses were performed operating in negative mode. The nitrogen flow rate was set at 15.0 mL min^−1^ and maintained at 350 °C, whereas the capillary voltage was set at 1.5 kV. Analyses were performed in triplicate. The different compounds were identified by comparing the retention time (RT), UV–Vis, and MS/MS spectra with reference compounds. Finally, the quantification was performed using external calibration curves (range: 0.03–0.15 µg/mL). The results were expressed as µg g^−1^ FW.

### 2.8. Statistical Analysis

The data obtained from qPCR, HPLC, and biochemical assays were treated by using Systat 10. Mean value was calculated along with the SD. Paired *t*-test and Bonferroni-adjusted probability were used to assess the difference between treatments and controls. Processing and statistical analysis of the microarray data were performed in R using Bioconductor package limma [27]. The raw microarray data were subjected to background subtraction and loess-normalized. Agilent control probes were filtered out. The linear models implemented in limma were used for finding DEGs. Comparisons were made for each of the treatment. Benjamini and Hochberg (BH) multiple testing correction was applied [28]. Heatmaps were obtained with Heatmapper (http://www.heatmapper.ca/ accessed on 1 September 2022) [29] by using Pearson clustering with single linkage method.

## 3. Results

### 3.1. Transcriptomic Analyses Reveal Differential Root/Shoot DEGs of Oxidative Stress-Related Activity in Plants Exposed to NNMF

*A. thaliana* seedlings vertically grown in Petri dishes and exposed to NNMF conditions from 10 min to 96 h were separately sampled by collecting roots and shoots. Controls plants were grown in the same experimental conditions (i.e., temperature, gravity, atmospheric pressure, and photosynthetic phlux density—PPD) but under GMF. Samples were assayed by microarray gene expression. Appendix A collects all information on the gene expression fold change volcano plots for all evaluated time points. Gene microarray data were then filtered by considering all genes coding for enzymes involved in oxidative reactions. A total of 194 DEGs were selected. A consistent percentage of these DEGs explained a fold change value >2 at almost all time points (Appendix A). In the following text, the term “not regulated” is referred to genes that are DEGs with fold change values below the established threshold limits of >2 and <0.5.

In general, the gene ontology (GO) analysis of the selected genes showed that the cellular component DEGs were mainly expressed in the extracellular region, followed by cytoplasm, cell wall, and chloroplast-related DEGs (Figure 1A), with a molecular function mainly identified in catalytic activity and binding (Figure 1B). The biological process was associated to DEGs responding to stress, and other cellular and metabolic processes (Figure 1C). The data matrix of Appendix A was then subjected to heatmapping in order to identify specific root/shoot patterns of DEGs.

Figure 2 shows the heatmap depicting the time-course DEGs of roots and shoots in plants grown under NNMF with respect to GMF (this heatmap includes all DEGs). Based on the expression matrix, we selected five groups characterized by specific expression patterns (groups A–E), which were then organized in tables by selecting DEGs showing a fold change ≥2 or ≤0.5 in at least one time point (Table 1, Table 2, Table 3, Table 4 and Table 5). We identified three main timing of magnetic induction: (i) early (10 min to 2 h), (ii) intermediate (4 and 24 h), and (iii) late (48 and 96 h).

Table 1 lists the DEGs of group A (Figure 2) showing a fold change ≥2 or ≤0.5 in at least one time point that are not regulated or downregulated in roots and upregulated in shoots. Most of the DEGs in this group have a subcellular prediction in the extracellular space and belong to the peroxidase family proteins, followed by laccases and dioxygenases (Appendix A). In particular, several shoot peroxidases (*At1g34510*, *At2g43480*, *At5g24070*, *At1g05240*, *At4g26010*, *At5g67400*) were upregulated at early (10 min and 1 h) times, whereas the peroxidases *At1g30870*, *At1g49570*, and *At1g05250* were upregulated in intermediate (4 h) and late (48–96 h) times. In shoots, *At1g48700*, a 2-oxoglutarate (2OG) and Fe(II)-dependent oxygenase, was upregulated at almost all times and downregulated in roots at 48 h, whereas three other 2OG-Fe(II) oxygenases were upregulated at early (*At5g51930*), middle and late times (*At1g55290*) and at early and late times (*At4g25310*). In shoots, a gene that encodes a protein that is similar to laccase-like polyphenol oxidases (*At5g48100*) was upregulated at very early times, whereas a putative laccase (*At5g01050*) was upregulated at intermediate and late times. A gene coding for a multicopper oxidase type I family protein (*At1g21860*) was downregulated at early times in roots and upregulated at almost all times in shoots, whereas a gene with a similar function (*At1g21860*) was only upregulated in shoots at intermediate times (Table 1).

Table 2 lists the DEGs of group B (Figure 2) that are upregulated in shoots mostly at all times and upregulated in roots mainly at earlier and intermediate times. Even in this group, some DEGs had an extracellular prediction but many were cytosolic. The gene function was dominated by copper ion binding oxidoreductases, dioxygenases, FAD-binding oxidases, and a few peroxidases (Appendix A). Four DEGs coding for 2OG-Fe(II) oxygenases (*At1g28030*, *At1g52790*, *At2g44800*, and *At3g28490*), two for gibberellin dioxygenases (*At1g60980* and *At2g34555*), and four DEGs coding for oxidoreductases with copper ion binding activity (*At4g28090*, *At1g55570*, *At1g75790*, and *At1g21850*) were upregulated in shoots at all times and upregulated in roots at early/intermediate times. Five DEGs coding for oxidoreductases with a FAD-binding domain (*At1g11770*, *At4g20800*, *At2g38960*, *At1g32300*, and *At5g45180*) were always upregulated in shoots, whereas in roots upregulation was absent at late times and the same pattern was observed for three peroxidases (*At1g65990*, *At1g24110*, and *At3g42570.1*) (Appendix A).

DEGs of group C (Figure 2) with a fold change ≥2 or ≤0.5 are listed in Table 3. Group C contains DEGs that are highly upregulated in shoots at 24 h (Figure 1). Two cytoplasmic DEGs encoding for oxidoreductases with a FAD-binding domain were both upregulated in shoots at 24 h, but one of them (*At1g26410.1*) was downregulated in roots at earlier time stages, whereas the other (*At1g26390.1*) was upregulated at earlier times both in roots and shoots. A gene coding for a peroxidase (*At5g05340.1*) with extracellular localization and a chloroplastic methionine sulfoxide reductase (*At4g21830.1*), which respond to singlet oxygen, were only upregulated in shoots at 24 h (Appendix A).

Group D contains DEGs that are highly upregulated in shoots at 48 h (Figure 2). DEGs of this group with a fold change ≥2 or ≤0.5 are listed in Table 4. A gene localized in the cytoplasm coding for a jasmonate-induced dioxygenase (*At2g38240.1*) was only upregulated in shoots at 48 h whereas another cytosolic gene coding for a galactose-oxidase (*At3g57620.1*) showed a high upregulation at 48 h also in roots. A gene coding for a plasma membrane-located ferric-chelate reductase (*At5g23990.1*) was upregulated in shoots at almost all times and upregulated in roots at 48 h, whereas a gene coding for an extracellular peroxidase (*At3g50990.1*) was highly upregulated in shoots at 48 h and was also upregulated in roots at intermediate and late times (Appendix A).

The last group, E, is made by DEGs that are downregulated in shoots at 24 h (Figure 2). The only gene with a fold change ≥2 or ≤0.5 (Table 5) is coding for a putative 2-aminoethanethiol dioxygenase (*At1g18490.1*) (Appendix A).

### 3.2. Validation by Real-Time PCR Confirms Gene Microarray Expression Data

Three genes were selected in order to validate the general gene expression of microarray results and were tested at 96 h. The pattern of fold change of *At2g18140*, *At1g32300*, and *At3g56350* (Table 6) was in line with the gene expression reported in the gene microarray experiments (see Appendix A).

### 3.3. GMF Induces a Significant ROS Production

In order to evaluate the ROS production in NNMF conditions, we measured the H_2_O_2_ content during the middle and late stages of plant growth (24 h, 48 h, and 96 h). For technical reasons, measurements at early times were not possible due to the tiny amount of material. Analyses were performed on both roots and shoots of plants grown in GMF and NNMF.

In general, the content of H_2_O_2_ was reduced 5-fold at 24 h, 2-fold at 48 h, and 0.2-fold at 24 h in plants grown under NNMF in comparison to GMF condition, independently of the analyzed plant tissue (Table 7).

Under GMF, the shoot H_2_O_2_ content decreased with time, whereas under NNMF, the H_2_O_2_ concentration was practically unaffected at 24 and 48 h and significantly decreased only at 96 h (Table 7). By calculating the NNMF/GMF ratio, a general increasing trend was observed for both shoots and roots. In shoots, NNMF always showed highly significantly (*p* < 0.01) lower H_2_O_2_ production with respect to GMF (Table 7).

Under GMF, root H_2_O_2_ production was almost constant with time, whereas a significant (*p* < 0.05) increase in H_2_O_2_ was observed with time in roots exposed to NNMF (Table 7). However, as for shoots, the NNMF/GMF ratio indicated a general reduction of root H_2_O_2_ in NNMF exposed plants (Table 7).

Because the observed trend could be related to different DEGs involved in H_2_O_2_ metabolism, DEGs involved in H_2_O_2_ metabolic processes were extracted from Appendix A. These genes were then used to generate a heatmap calculated using a Pearson distance and average linkage method (Figure 3).

DEGs (NNMF/GMF fold change ratio) were considered by analyzing data from 4 h to 48 h in order to evaluate the possible regulation before the first (24 h) and last (96 h) time considered for ROS scavenging/production (Table 7). In general, the heatmap confirms the DEGs regulation of roots and shoots ROS-scavenging enzymes. In particular, roots showed a marked downregulation of most of DEGs responsible for the transcription of ROS-scavenging enzymes. The only exception was for *RBOHH* and *RBOHG* at 4 h and some peroxidases at 4 h (*PRX71*, *At1g77100*, *At5g64110*, *At4g33870*, *At5g58400*), 24 h (*PRX71*, *At1g77100*, *At5g24070*) and 48 h (*At1g77100*, *At5g19880*, *At4g17690*, *At2g18150*), which were upregulated (Figure 3). These data are in agreement with the general time-course increasing H_2_O_2_ levels in roots exposed to NNMF (Table 7). On the other hand, shoot upregulation of most of ROS-scavenging DEGs was evident (Figure 3), with stronger upregulation present at 4 h for *CAT1* (*At1g20630*), *APX2* (*At3g09640*), *PRX2* (*At1g05250*), and *PER57* (*At5g17820*); at 24 h for *APX1* (*At1g07890*), *CCS* (*At1g12520*), *SOD1* (*At1g08830*), and *SOD2* (*At2g28190*); and at 48 h for *RBOHJ* (*At3g45810*), *RBOHG* (*At4g25090*), and *PER36* (*At3g50990*). Again, DEGs data obtained by microarray were in agreement with the decreasing shoot H_2_O_2_ production with time in NNMF exposed plants (Table 7).

### 3.4. Reduction of GMF to NNMF Modulates the Production of Antioxidant Polyphenols

HPLC-DAD-MS/MS analyses were performed on the same extracts used for H_2_O_2_ detection. The considered time points were then 24 h, 48 h, and 96 h for both shoots and roots. In general, 44 compounds belonging to different polyphenol classes were identified in both NNMF and GMF samples, including 8 flavanols (Figure 4 compounds **7**, **11**, **16**, **19**, **26**, **34**, **41**, and **45**), 13 flavonols (compounds **10**, **13**, **14**, **15**, **18**, **20**, **22**, **24**, **29**, **33**, **39**, **42**, and **43**), 10 isoflavanones (compounds **3**, **5**, **6**, **9**, **21**, **28**, **31**, **35**, **36**, and **40**), and 6 *O*-methylated flavonols (compounds **12**, **17**, **30**, **32**, **44**, and **23**) (see Appendix A for quantitative variation and Figure 4 for chemical structures).

Table 8 shows the content of the total identified polyphenols expressed as µg g^−1^ FW, along with the NNMF/GMF ratio. In general, at 96 h, both roots and shoots showed a total polyphenol content that was always lower under NNMF than GMF. From 24 h to 48 h, the shoots polyphenol content showed a decreasing trend with time, whereas in roots the highest value was found at 48 h (Table 8).

**Figure 4 biomolecules-12-01824-f004:**
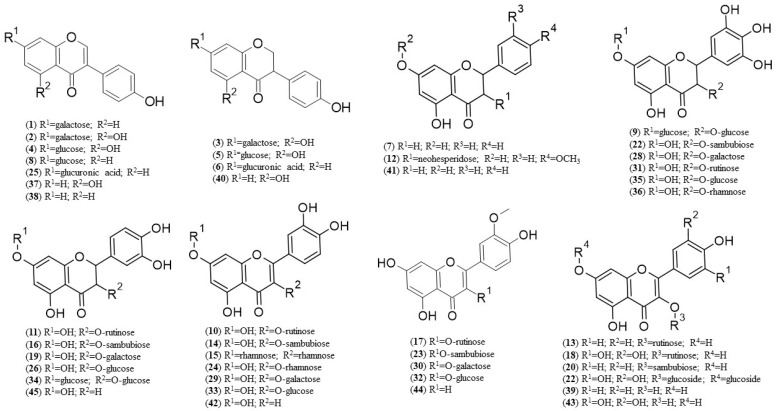
Chemical structure of polyphenolic compounds identified in *Arabidopsis thaliana* roots and shoots grown under GMF and NNMF conditions. Formulae numbers refer to the compounds cited in the main text and in Table 9.

In order to evaluate whether NNMF could affect the content of specific polyphenols, raw data obtained from HPLC-DAD-MS/MS analyses (see Appendix A) were used to calculate the NNMF/GMF ratio (Table 9). Data indicate that NNMF differently modulates the content of polyphenols, not only within the two organs, but also with time.

In roots, plants grown for 24 h under NNMF conditions showed a general increase in polyphenols, when compared to GMF, with the sole exception for isorhamnetin glucoside (**32**) and quercetin glucoside (**33**) (Table 9). At 48 h, the production of polyphenols was even higher in NNMF, with the exception for dihydroquercetin-rutinoside (**11**) and isorhamnetin (**44**), which were reduced. At 96 h, root polyphenols showed a reversed trend and only daidzein-galactoside (**1**), dihydrogenistein-glucuronide (**6**), isorhamnetin-sambubioside (**23**), dihydroquercetin-glucoside (**26**), dihydromyricetin-glucoside (**35**) and genistein (**37**) remained upregulated; interestingly, dihydroquercetin-rutinoside (**11**), which was reduced at 24 h, increased at 96 h (Table 9).

At all times, shoots contained 14 polyphenols that were not found in roots: genistein-galactoside (**2**), dihydrogenistein-galactoside (**3**), genistein-glucoside (**4**), dihydrogenistein-glucoside (**5**), dihydrokaempferol-glucoside (**7**), dihydrohesperetin-neohesperidoside (**12**), kaempferol-sambubioside (**20**), dihydromyricetin-sambubioside (**28**), myricetin-diglucoside (**22**), dihydromyricetin-rutinoside (**31**), dihydroquercetin-diglucoside (**34**), kaempferol (**39**), dihydrokaempferol (**41**) and dihydroquercetin (**45**). However, shoots lacked isorhamnetin-galactoside (**30**) and isorhamnetin-glucoside (**32**) at all times and kaempferol-rutinoside (**13**) at 24 h, with respect to roots. At 24 h, most of the polyphenols were upregulated in NNMF, with particular reference to dihydrogenistein-galactoside (**3**); however, a significant reduction was found for genistein-galactoside (**2**), dihydrogenistein-glucuronide (**6**), dihydroquercetin-rutinoside (**11**), daidzein-glucuronide (**25**), dihydromyricetin-galactoside (**28**), quercetin-glucoside (**33**), and an unknown polyphenol (**27**). Upregulation was also found at 48 h NNMF treated plants for most of the identified compounds, with a strong increase of dihydrokaempferol (**41**); whereas dihydrogenistein-galactoside (**3**) and dihydroquercetin-glucoside (**26**) were severely reduced. Finally, at 96 h a general reduction of most of the shoot polyphenols was observed in NNMF plants, with the sole exception for myricetin-diglucoside (**22**), dihydrokaempferol (**41**), and an unknown polyphenol (**27**) that showed a higher induction with respect to GMF plants (Table 9).

In order to evaluate whether the observed differences in the root and shoot polyphenols could match the DEGs, we extracted from the general gene microarray analysis those genes that were involved in flavonoid biosynthesis (see Appendix A). Data were then used, along with those reported in Table 9, to generate heatmaps coupled with hierarchical cluster analysis (Figure 5).

A direct comparison between polyphenol quantity (Figure 5A) and DEGs expressing polyphenol metabolism (Figure 5B) shows a gene upregulation of *chalcone synthase* (*At5g13930*), *chalcone isomerase* (*At5g05270*) and *UDP-glucosyl transferase* (*At5g17040*) in roots at 4 h, that preceded the reduced biosynthesis of isorhamnetin-galactoside (**30**), daidzein (**38**), quercetin-dirhamnopyranoside (**15**), kaempferol-rutinoside (**13**), dihydrogenistein (**40**), and isorhamnetin-sambubioside (**23**) that were detected at 24 h (Figure 5 green arrows). At 24 h, root upregulation of *phenylcoumaran benzylic ether reductase 1* (*At4g39230*), *pinoresinol reductase 1* (*At1g32100*), *pinoresinol reductase 2* (*At4g13660*), and *polyketide synthase A* (*At1g02050*) preceded the biosynthesis of several polyphenols at 48 h, with particular reference to quercetin-glucoside (**33**), myricetin (**43**), quercetin-galactoside (**29**), and dihydrogenistein (**40**) (Figure 5 orange arrows). Finally, the root DEGs at 48 h were characterized by general downregulation, which matched with the low amount of polyphenols detected at 96 h (Figure 4A), with the exception for *DFR* (*At5g42800*), *EPSPS* (*At1g48860*), *CYP75B1* (*At5g07990*), and a putative *isoflavone reductase* (*At1g75290*), which upregulation-matched with the higher levels of dihydrogenistein-glucuronide (**6**), dihydroquercetin-glucoside (**26**), and dihydromyricetin-glucoside (**35**) (Figure 5 blue arrows).

With regards shoots, the gene upregulation of chalcone synthase (At4g00040 and At1g02050), chalcone isomerase (At3g55120), flavonol synthase (At5g63580), and UDP-glucosyl transferase (At5g17040 and At2g18560) preceded the synthesis of the most polyphenols detected at 24 h, including genistein (**37**), quercetin-sambubioside (**14**), quercetin-glucoside (**33**), quercetin-galactoside (**29**), kaempferol (**39**), and kaempferol-rutinoside (**13**) (Figure 5 purple arrows). The shoot DEGs at 24 h were characterized by a downregulation of several flavonol synthases including FLS3 (At5g63590), FLS4 (At5g63595), and FLS5 (At5g63600) that matched with the reduced production of dihydrogenistein-galactoside (**3**), daidzein-glucuronide (**25**), and quercetin-galactoside (**29**) at 48 h (Figure 3 black arrows). The upregulation of UDP-glucosyl transferase (At5g17030), FLS6 (At5g43935), and a putative isoflavone reductase (At1g75290) at 24 h matched with the increased amount, among others, of dihydrogenistein-glucuronide (**6**), dihydroquercetin-rutinoside (**11**), quercetin-dirhamnopyranoside (**15**), dihydromyricetin-sambubioside (**21**), and dihydrokaempferol (**41**) (Figure 3 maroon arrows). Finally, the shoot DEGs at 48 h were characterized by the upregulation of an isoflavone reductase (At1g75290), FLS6 (At5g43935), chalcone and stilbene synthase (At5g66220), UDP-glucosyl transferase (At5g17030), DFR (At5g42800), and BAN (At1g61720), that matched the higher production of myricetin-diglucoside (**22**), an unknown polyphenol (**27**), and dihydrokaempferol (**41**) at 96 h (Figure 5 light blue arrow).

## 4. Discussion

MF variations induce in plants a typical abiotic stress response, with differential expression of several genes, including DEGs coding for responses to oxidative stress [20]. ROS comprise a wide array of oxygen-derived compounds with varying rates of reactivity and may be produced by either endogenous or exogenous stimulation [30]. H_2_O_2_, a non-radical ROS, is mainly produced by the plasma membrane-located NADPH oxidase (also known as the respiratory burst oxidase homologs, RBOHs) that produces the superoxide anion, which in turn is reduced by superoxide dismutase (SOD) to H_2_O_2_ [31]. The latter is detoxified to water by a number of scavenging enzymes, including several peroxidases [32]. Despite the disrupting effects, the timely and appropriate production of H_2_O_2_ can act as a signal for plant responses to environmental stresses [33] and our data indicate that in normal GMF conditions ROS are produced.

Evidence indicates that reduction in GMF affects cellular ROS levels and alters their normal signaling roles in regulating the diverse cellular physiological processes [34]. The effect of both static MF and extremely low-frequency MF have been demonstrated to exert an effect on ROS production [3,4,21] and our data provide evidence on the relationship between H_2_O_2_ production and metabolomics/transcriptomics of ROS scavenging systems under NNMF conditions.

Our results show that reduction of GMF to NNMF causes root/shoot expression of several DEGs coding for enzymes involved in oxidative reactions and confirm the hormetic behavior in time-course experiments. We identified several DEGs in the two analyzed organs (roots and shoots). We noted the constant upregulation of shoot *monoamine-oxidase A repressor R1 protein* (*At1g67270*), which is always downregulated in roots. Because monoamine oxidase A (MAO-A) degrades biogenic amines producing reactive oxygen, the inhibition of MAO-A promoter by the upregulation of this zinc-finger transcription factor may result in the reduction not only of MAO enzymatic activity, but also in the concomitant decrease of ROS production in the shoots [35]. A common root and shoot upregulation was found for several oxygenases. Among these, *AERO2* is known for its involvement in the oxidative protein folding in the endoplasmic reticulum [36], whereas *GulLO1*, which takes also part as *AERO2* of the secretory pathway, is known to code for a d-arabinono-1,4-lactone oxidase and is involved in the production of L-ascorbic acid, an important antioxidant and redox molecule [37]. Other common DEGs were *GOXL5*, a galactose oxidase that is involved in pectin crosslinking through modification of galactose side chain [38], *OGOX3*, an oligogalacturonide oxidase plant berberine bridge enzyme-like oligosaccharide oxidase that oxidizes oligogalacturonide and cellodextrin [39], and several 2-oxoglutarates (*2OG*) and Fe(II)-dependent oxygenases, which are important oxidizing biological catalysts whose activity is usually increased by the contribution of catalase and ascorbate [40]. Interestingly, a consistent downregulation was found for a plant cysteine oxidase (*PCO3*), which is involved in the N-end rule pathway-mediated proteolysis for degradation via the 26S proteasome [41].

The reduction of GMF to NNMF also decreased the *A. thaliana* H_2_O_2_ production both in roots and shoots. This effect appears to be related to a modulation of DEGs coding for enzymes involved in ROS scavenging as well as the increased production of polyphenols with antioxidant properties. As already observed in our previous studies [20], the differential gene expression between organs suggests the regulation of overlapping sets of stress-responsive genes.

Both NNMF and GMF plants show developmental changes in ROS production. NNMF exposed plants show an increasing trend in ROS production, although these values are always lower when compared to GMF plants (see Table 7). However, this increase was associated to the upregulation of genes encoding for enzymes involved in ROS production, such as *RBOHH*, *RBOHJ*, *RBOHG*, *SOD1*, *SOD2*, and *CCS*. RBOHH and RBOHJ are regulated by cytosolic calcium and protein phosphorylation, and their H_2_O_2_ production was found to dampen growth rate oscillations [42]. Previous studies have shown that NNMF modulates the ion homeostasis and affects calcium levels [43]. Although no data are available on RBOHG function, some RBOHs have been found to activate a ROS signaling that induces polyphenol biosynthesis genes, thus enabling their accumulation in plants [44]. SOD1 is cytosolic and SOD2 is attached to thylakoid membranes where PSI is located and are two isoforms of SOD expressed in photosynthetic tissues [45]. Overexpression of SOD genes increases the tolerance to abiotic stresses in many plant species, including *A. thaliana* [46]. CCS, or Cu chaperone for SOD (*At1g12520*), is localized to the chloroplast, expressed in roots and shoots, and is required to insert Cu and activate SOD [47]. SOD1 activation in the cytoplasm involves both a CCS-dependent and -independent pathways, whereas SOD2 dismutates the superoxide anion produced by PSI, resulting in the formation H_2_O_2_, and its activation depends totally on CCS [48].

We also noticed that the reduction in H_2_O_2_ of NNMF plants was associated with the upregulation of several genes coding for ROS scavengers, including several peroxidases, some of which (*PER6*, *PER28*, *PER36*, and *At3g42570*) were upregulated at all sampling times and in both organs. These class III peroxidases (E.C.1.11.1.7) are involved in oxidation reactions that use H_2_O_2_ as an electron acceptor and several substrates as electron donors [49]. PER6, PER28, and the peroxidase coded by *At3g42570* are present in all organs [50], whereas PER36 is extracellular and mainly located in internodes and developing seeds. PER6 loosens cell walls, apparently contributing to disruption of polysaccharide bonds, which is required for cell elongation during growth [49].

There was a clear relationship between H_2_O_2_ production and the presence of antioxidant biomolecules. Even in this case, polyphenols were observed to change their content during development in both NNMF- and GMF-exposed plants. The increasing trend of H_2_O_2_ production in NNMF-exposed plants was associated with a progressive reduction of polyphenols (Table 7 and Table 8). Since antioxidant polyphenols scavenge free radicals and repair damaged cells, they are active in protecting plant cells from oxidative stress [51]. At 24 h of NNMF exposure, in both roots and shoots, the high content of polyphenols was dominated by flavonols, isoflavones, and isoflavanones. Interestingly, the high production of the isoflavonoids daidzein (**38**) and dihydrogenistein (**40**) was preceded by the upregulation of an isoflavone reductase (*At1g75290*), whereas the higher contents of the flavonols dihydroquercetin-glucoside (**26**), quercetin-sambubioside (**14**), dyhydroquercetin-sambubioside (**16**), and isorhamnetin-sambubioside (**23**) were preceded by the upregulation of genes involved in flavan-3-ols synthesis (*BAN*) and flavonol biosynthetic process (*FLS6*). The formation of flavonols depends on flavonol synthase (FLS); in *A. thaliana*, only *FLS1* appears to be functionally expressed and *FLS6* has been identified as pseudogene because of its considerably truncated C-termini [52]. Nevertheless, some FLS might retain some activity in flavonol metabolism [53].

Low levels of polyphenols in both roots and shoots at 96 h NNMF exposure were associated to a general drop in flavonol biosynthesis, which corresponded to the downregulation of *FLS2* and *DRL1*. *FLS2* is present in small and medium-size leaves and its expression has been reported to be constitutive but much lower than for *FLS1* [54]. *DRL1* is essential for pollen development and male fertility in *A. thaliana* and encodes for a dihydroflavonol 4-reductase-like1 protein, which is an NADPH-dependent enzyme converting dihydroflavonols to their corresponding leucoanthocyanidins [55]. Among polyphenols, both roots and shoots of NNMF plants show a significant drop of quercetin-rutinoside (**10**), isorhamnetin-rutinoside (**17**), and dihydromyricetin-galactoside (**28**).

## 5. Conclusions

Either static or oscillating weak magnetic fields with a higher intensity than the GMF have been reported to induce oxidative stress [6,56], suggesting the involvement of the radical pair mechanisms [15] with the contribution of both mitochondria [57] and chloroplasts [58]. The primary effect of a MF with respect to cryptochrome function has been postulated to be the modulation of ROS [19,59]. This mechanism perfectly predicts an effect on cellular ROS signaling pathways; therefore, this hypothesis explains the ROS-related modulation of gene expression in response to reduced MFs as observed in our work.

Typically, variations in MF-dependent ROS production are accompanied by variations in antioxidants [60]. From a transcriptomic point of view, we identified different classes of “oxidases” including RBOHs and several peroxidases that play hormetic roles in both root and shoot MF-dependent oxidative stress. Our results also indicate that polyphenols represent a contributing factor involved in the antioxidant response to varying MFs and confirm previous studies indicating the modulation of flavonoids and other polyphenols by MF [61,62,63,64]. We identified some key isoflavonoids like daidzein (**38**) and dihydrogenistein (**40**), and flavonols including dihydroquercetin-glucoside (**26**), quercetin-sambubioside (**14**), dyhydroquercetin-sambubioside (**16**), and isorhamnetin-sambubioside (**23**), along with the regulation of the associated genes such as isoflavone reductase (*At1g75290*) and some flavonol synthases that deserve further studies.

In conclusion, our results indicate that the GMF induces a basic oxidative stress, which is characterized by the downregulation of genes coding for scavenging enzymes and the upregulation of genes that code for enzymes that contribute to ROS production. We hypothesize that this condition generates a mild oxidative stress condition that plants evolved in a GMF environment. Changes in MF induce in plants changes in the redox status, indicating a functional role of plant magnetoperception in response to stress. Therefore, we believe that MF variations can be considered an abiotic stress factor, triggered by a still-unknown mechanism of magnetic induction [18] that is first evidenced by a plant’s alteration of the redox status.

## Figures and Tables

**Figure 1 biomolecules-12-01824-f001:**
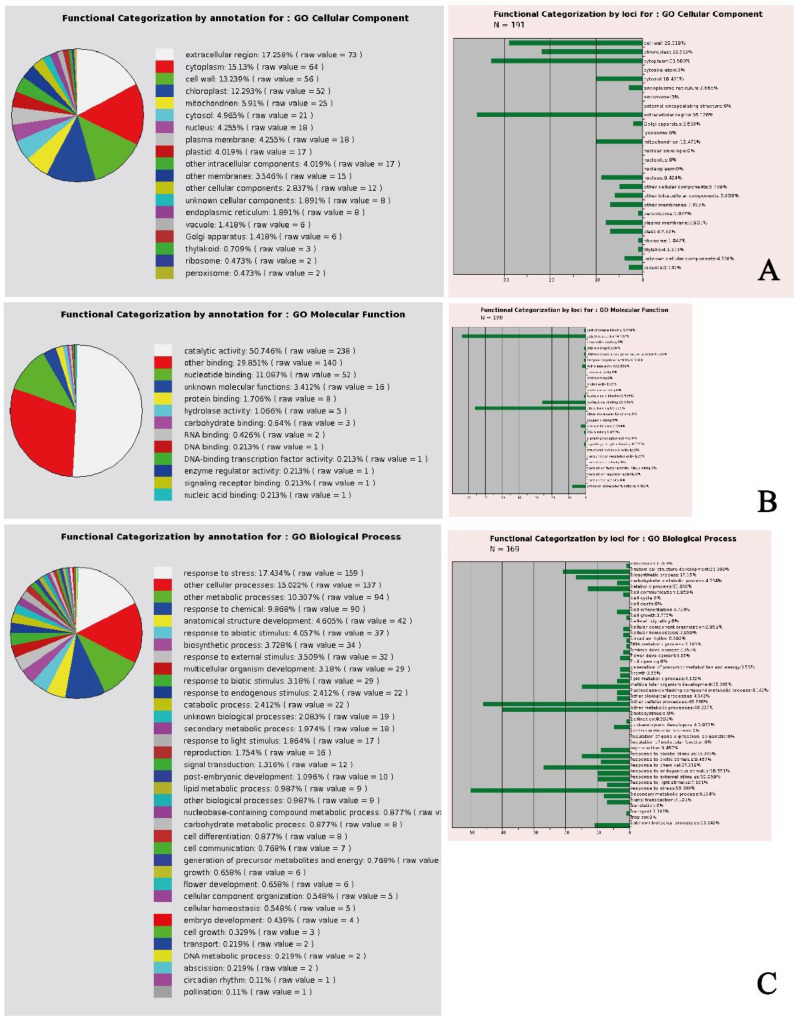
Gene ontology analysis of *Arabidopsis thaliana* genes selected for their involvement in oxidative stress. (**A**) Cellular component; (**B**) molecular function; (**C**) biological process. See text for further explanation.

**Figure 2 biomolecules-12-01824-f002:**
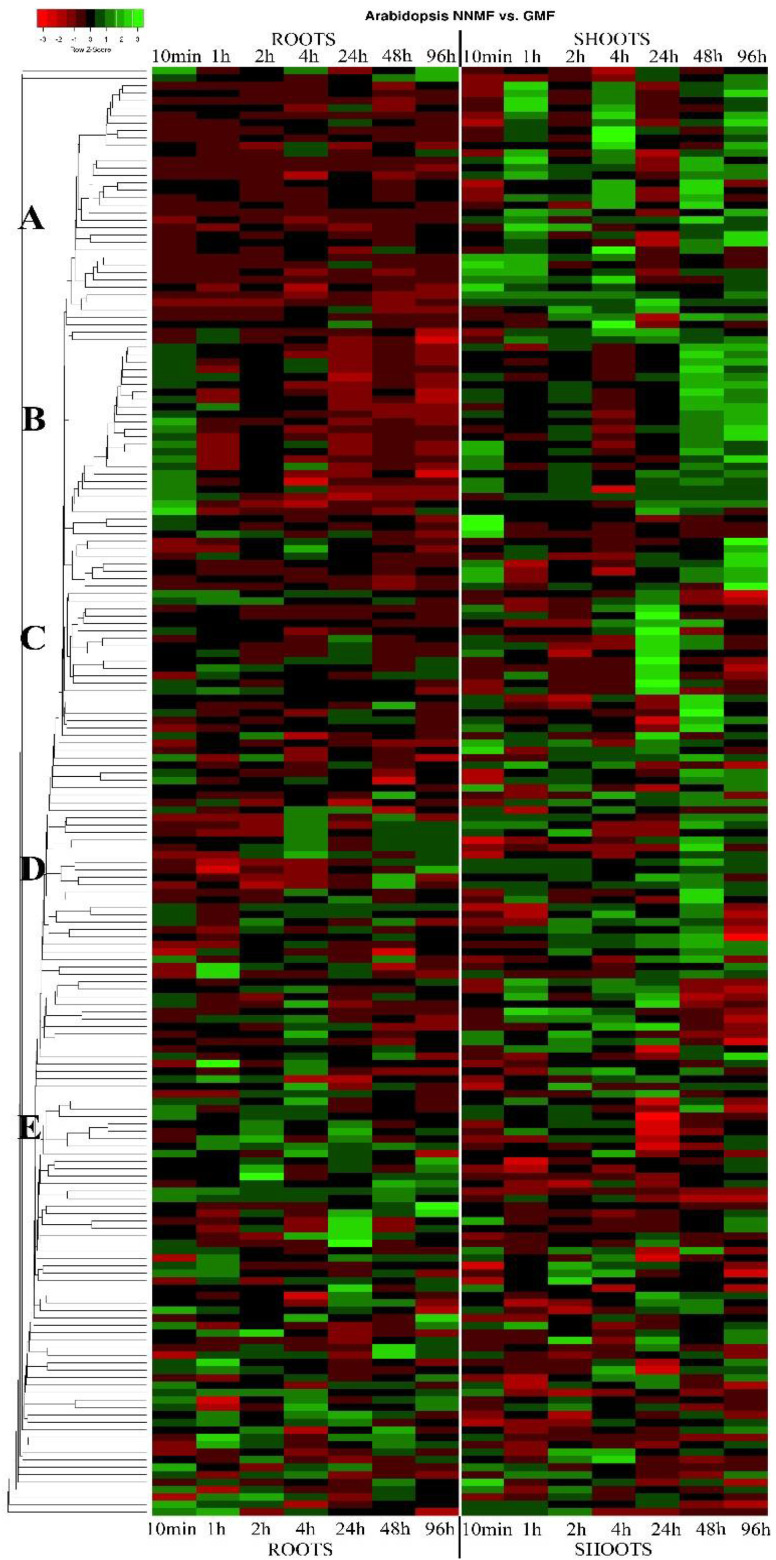
Heatmap of genes coding for enzymes involved in oxidation reactions from the time-course microarray gene expression of *A. thaliana* grown in a near-null magnetic field (NNMF) compared to geomagnetic field (GMF) grown plants. The heatmap was generated by using a single linkage clustering method and Pearson distance measuring method. Downregulated genes are marked in red-shaded colors, whereas upregulation is evidenced by different shades of green. Five groups of genes are evidenced (see text for more details).

**Figure 3 biomolecules-12-01824-f003:**
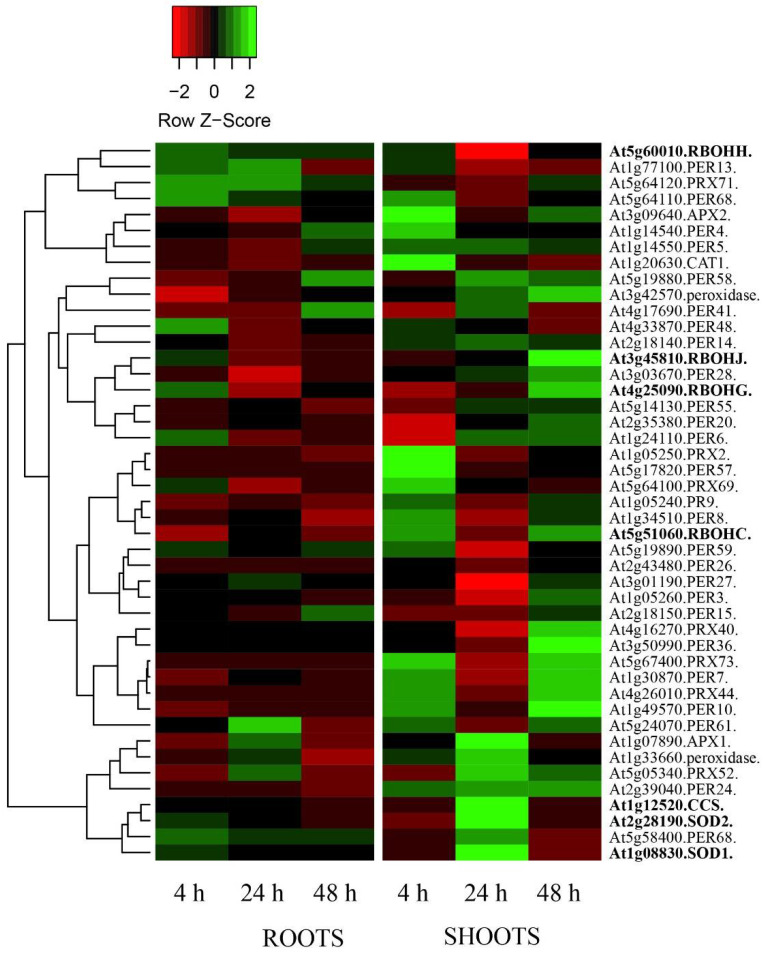
Gene expression fold change heatmap calculated with a Pearson distance and average linkage method from genes coding for ROS-scavenging enzymes selected from Appendix A. The heatmap was generated by using a single linkage clustering method and Pearson distance measuring method. Gene coding for ROS-producing enzymes are evidenced in boldface. Downregulated genes are marked in red-shaded colors, whereas upregulation is evidenced by different shades of green.

**Figure 5 biomolecules-12-01824-f005:**
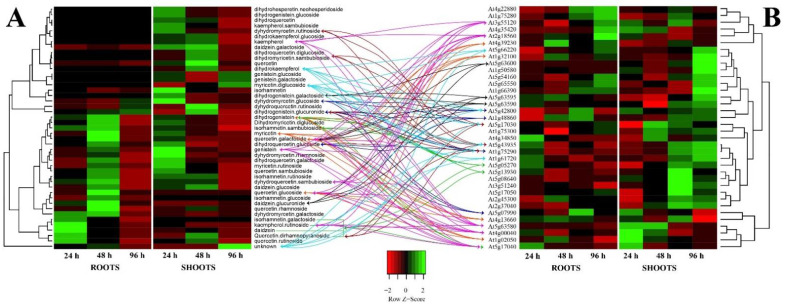
Direct comparison of heatmaps of polyphenols fold change obtained from data of Table 9 (Panel (**A**)) and fold change of selected genes involved in polyphenols’ metabolisms obtained from Appendix A (Panel (**B**)). The heatmap was generated by using a single linkage clustering method and a Pearson distance measuring method. Downregulated genes are marked in red-shaded colors, whereas upregulation is evidenced by different shades of green. See text for arrow color specifications.

**Table 1 biomolecules-12-01824-t001:** Group A. Genes not regulated or downregulated in roots and upregulated in shoots. Genes were selected based on a fold change ≥2 or ≤0.5 in at least one time point. Values are expressed as the mean and standard deviation (±), values with a fold change ≥2 or ≤0.5 are indicated as boldfaced numbers.

		ROOT							SHOOT						
Gene Locus	Gene Code	10 min	1 h	2 h	4 h	24 h	48 h	96 h	10 min	1 h	2 h	4 h	24 h	48 h	96 h
At4g37160	SKS15	0.75 ± 0.16	0.77 ± 0.08	0.93 ± 0.08	0.82 ± 0.11	0.92 ± 0.21	0.76 ± 0.18	0.67 ± 0.03	0.72 ± 0.11	1.16 ± 0.54	0.58 ± 0.4	**2.00 ± 0.65**	1.03 ± 0.22	0.99 ± 0.38	1.10 ± 0.61
At1g21860	SKS7	**0.50 ± 0.10**	1.78 ± 0.43	0.86 ± 0.44	0.63 ± 0.11	0.79 ± 0.44	0.94 ± 0.60	0.92 ± 0.19	**2.11 ± 0.34**	**2.50 ± 0.77**	0.98 ± 0.69	**2.12 ± 0.24**	1.84 ± 0.31	**3.62 ± 2.64**	1.98 ± 0.25
At4g25310	2OG and Fe(II)-dependent oxygenase	0.88 ± 0.11	1.14 ± 0.13	1.11 ± 0.13	1.07 ± 0.12	0.95 ± 0.06	1.12 ± 0.20	0.83 ± 0.47	1.56 ± 0.31	**2.39 ± 0.50**	1.72 ± 0.16	1.77 ± 0.31	0.72 ± 0.03	**2.72 ± 0.56**	**2.32 ± 1.92**
At1g55290	F6′H2	1.33 ± 0.13	1.13 ± 0.04	0.99 ± 0.10	1.15 ± 0.07	1.39 ± 0.21	0.88 ± 0.48	1.23 ± 0.14	0.50 ± 0.03	1.38 ± 0.85	1.25 ± 0.25	**2.17 ± 1.33**	1.15 ± 0.26	**2.40 ± 2.07**	0.78 ± 0.20
At5g51930	GMC	0.84 ± 0.23	0.82 ± 0.05	0.78 ± 0.12	1.08 ± 0.18	0.85 ± 0.16	0.79 ± 0.13	0.78 ± 0.08	**2.30 ± 1.96**	**2.40 ± 0.44**	1.45 ± 0.63	1.79 ± 0.12	0.88 ± 0.31	1.32 ± 0.34	0.72 ± 0.08
At1g19900	RUBY	1.00 ± 0.10	0.98 ± 0.11	0.93 ± 0.1	0.96 ± 0.12	1.51 ± 0.84	0.67 ± 0.43	0.75 ± 0.06	1.05 ± 0.41	0.62 ± 0.22	1.02 ± 0.26	**2.49 ± 1.32**	0.55 ± 0.37	0.92 ± 0.08	0.65 ± 0.21
At5g22410	RHS18	1.06 ± 0.22	0.99 ± 0.16	0.92 ± 0.05	0.81 ± 0.22	0.98 ± 0.20	0.57 ± 0.30	1.06 ± 0.20	0.95 ± 0.50	3.59 ± 2.18	1.04 ± 0.74	**2.53 ± 1.26**	1.09 ± 0.18	1.35 ± 0.78	**2.09 ± 1.57**
At5g48100	LAC15	1.28 ± 0.18	0.90 ± 0.15	1.34 ± 0.15	0.64 ± 0.16	1.37 ± 0.21	1.14 ± 0.23	0.96 ± 0.11	**2.25 ± 1.78**	1.45 ± 0.26	1.36 ± 1.05	1.50 ± 0.35	1.23 ± 0.78	1.00 ± 0.22	1.65 ± 0.53
At5g01050	LAC9	1.04 ± 0.19	1.08 ± 0.11	1.01 ± 0.06	1.23 ± 0.16	0.93 ± 0.06	1.11 ± 0.19	1.04 ± 0.18	0.90 ± 0.17	1.62 ± 0.63	1.21 ± 0.73	**2.97 ± 1.57**	1.72 ± 0.39	1.15 ± 0.08	**2.04 ± 0.50**
At1g48700	CP5	0.69 ± 0.06	0.61 ± 0.30	0.64 ± 0.32	0.82 ± 0.12	1.03 ± 0.19	**0.38 ± 0.09**	1.15 ± 0.83	**2.01 ± 0.33**	1.87 ± 0.39	**2.22 ± 0.40**	1.90 ± 0.09	**3.45 ± 0.96**	**2.20 ± 0.44**	1.94 ± 0.77
At1g05240	PR9	1.17 ± 0.22	1.19 ± 0.04	0.97 ± 0.05	0.95 ± 0.21	1.13 ± 0.11	0.97 ± 0.19	1.01 ± 0.25	0.61 ± 0.08	**2.02 ± 0.87**	1.06 ± 0.67	1.86 ± 1.34	0.92 ± 0.22	1.44 ± 0.18	**2.32 ± 0.75**
At1g05250	PRX2	0.95 ± 0.17	1.07 ± 0.03	0.89 ± 0.05	0.86 ± 0.21	1.01 ± 0.13	0.83 ± 0.16	0.98 ± 0.28	0.58 ± 0.07	1.79 ± 1.32	0.99 ± 0.75	**2.83 ± 1.93**	0.67 ± 0.17	1.23 ± 0.19	**2.18 ± 0.56**
At1g30870	PER7	1.02 ± 0.19	1.02 ± 0.07	0.93 ± 0.05	0.91 ± 0.19	1.27 ± 0.20	0.95 ± 0.24	1.02 ± 0.16	0.54 ± 0.10	1.10 ± 0.69	1.03 ± 0.73	**2.07 ± 1.07**	0.67 ± 0.08	**2.26 ± 2.10**	1.20 ± 0.14
At1g34510	PER8	1.16 ± 0.18	0.90 ± 0.08	1.04 ± 0.23	1.13 ± 0.08	1.53 ± 1.17	0.60 ± 0.49	1.23 ± 0.49	0.67 ± 0.12	**3.28 ± 2.67**	1.27 ± 0.58	**2.42 ± 2.35**	0.57 ± 0.29	1.93 ± 0.34	**2.67 ± 0.79**
At1g49570	PER10	0.72 ± 0.14	0.82 ± 0.13	1.03 ± 0.14	0.73 ± 0.22	0.99 ± 0.32	0.92 ± 0.21	0.57 ± 0.48	0.72 ± 0.13	0.86 ± 0.45	0.37 ± 0.27	1.69 ± 0.78	0.96 ± 0.06	**2.14 ± 1.31**	0.90 ± 0.55
At2g43480	PER26	0.85 ± 0.18	0.91 ± 0.09	0.92 ± 0.02	0.82 ± 0.04	0.88 ± 0.14	0.93 ± 0.06	0.82 ± 0.05	1.06 ± 0.30	**2.05 ± 0.04**	1.75 ± 0.34	1.10 ± 0.26	0.55 ± 0.14	1.05 ± 0.18	**2.09 ± 0.23**
At4g26010	PRX44	1.18 ± 0.22	1.10 ± 0.15	0.99 ± 0.03	0.99 ± 0.27	1.01 ± 0.16	0.98 ± 0.27	1.11 ± 0.10	1.59 ± 0.23	**2.66 ± 1.22**	1.39 ± 0.55	**2.06 ± 1.35**	0.77 ± 0.19	**2.33 ± 2.19**	1.31 ± 0.33
At5g24070	PER61	1.01 ± 0.1	1.14 ± 0.19	1.12 ± 0.11	1.13 ± 0.09	1.70 ± 0.78	0.84 ± 0.32	0.79 ± 0.11	**3.39 ± 4.57**	**3.23 ± 2.41**	0.85 ± 0.56	1.52 ± 0.06	0.90 ± 0.42	1.46 ± 0.54	1.12 ± 1.01
At5g67400	PRX73	1.03 ± 0.17	1.15 ± 0.20	1.01 ± 0.11	1.01 ± 0.19	1.21 ± 0.18	1.03 ± 0.28	1.23 ± 0.21	0.62 ± 0.03	**2.08 ± 0.68**	1.58 ± 0.83	**2.21 ± 1.38**	0.72 ± 0.07	**2.25 ± 1.88**	1.11 ± 0.20
At1g67270	Monoamine-oxidase A repressor R1 protein	0.96 ± 0.13	0.99 ± 0.08	0.93 ± 0.16	0.88 ± 0.31	0.92 ± 0.17	0.71 ± 0.09	0.70 ± 0.38	**2.14 ± 0.33**	**2.21 ± 0.12**	**2.32 ± 0.41**	**2.11 ± 0.25**	1.97 ± 0.54	1.69 ± 0.91	**2.34 ± 0.78**

**Table 2 biomolecules-12-01824-t002:** Group B. Genes that are upregulated in shoots mostly at all times and upregulated in roots mainly at earlier times. Genes were selected based on a fold change ≥2 or ≤0.5 in at least one time point. Values are expressed as the mean and standard deviation (±), values with a fold change ≥2 or ≤0.5 are indicated as boldfaced numbers.

		ROOT							SHOOT						
Gene Locus	Gene Code	10 min	1 h	2 h	4 h	24 h	48 h	96 h	10 min	1 h	2 h	4 h	24 h	48 h	96 h
*At3g59845*	*Zinc-binding dehydrogenase family protein*	**2.35 ± 0.30**	1.88 ± 0.15	**2.13 ± 0.19**	**2.49 ± 0.52**	1.89 ± 0.21	1.99 ± 0.30	1.91 ± 0.11	**2.38 ± 0.58**	**2.10 ± 0.11**	**2.28 ± 0.43**	2.07 ± 0.27	**2.21 ± 0.60**	**2.47 ± 0.36**	**2.27 ± 0.34**
*At4g28090*	*SKS10*	**2.50 ± 0.36**	1.88 ± 0.14	**2.14 ± 0.20**	**2.28 ± 0.31**	1.95 ± 0.20	**2.02 ± 0.33**	1.92 ± 0.11	**2.61 ± 0.32**	**2.14 ± 0.13**	**2.21 ± 0.49**	1.94 ± 0.08	**2.10 ± 0.52**	**2.40 ± 0.35**	**2.37 ± 0.36**
*At1g55570*	*SKS12*	**2.70 ± 0.49**	1.99 ± 0.11	**2.24 ± 0.21**	1.92 ± 0.23	**2.13 ± 0.20**	**2.13 ± 0.35**	**2.04 ± 0.11**	**2.05 ± 0.92**	**2.20 ± 0.18**	**2.45 ± 0.61**	1.87 ± 0.17	**2.51 ± 0.47**	**2.76 ± 1.93**	**3.35 ± 1.75**
*At1g75790*	*SKS18*	**3.03 ± 1.00**	1.91 ± 0.12	1.85 ± 0.29	1.61 ± 0.49	**2.03 ± 0.22**	1.89 ± 0.09	2.28 ± 0.45	**2.26 ± 0.38**	**2.27 ± 0.22**	**2.25 ± 0.51**	**2.09 ± 0.27**	**2.00 ± 0.86**	**2.89 ± 0.45**	**2.73 ± 0.68**
*At1g21850*	*SKS8*	**2.34 ± 0.29**	1.87 ± 0.15	**2.12 ± 0.19**	**2.09 ± 0.42**	1.89 ± 0.20	1.98 ± 0.30	1.72 ± 0.27	**2.35 ± 0.29**	**2.13 ± 0.13**	**2.27 ± 0.43**	**2.07 ± 0.27**	**2.21 ± 0.60**	**2.62 ± 0.44**	**2.53 ± 0.43**
*At1g32710*	*Cytochrome c oxidase*	**2.51 ± 0.35**	1.98 ± 0.12	**2.28 ± 0.27**	**2.03 ± 0.28**	**2.02 ± 0.20**	**2.13 ± 0.42**	**2.03 ± 0.11**	**2.31 ± 0.33**	**2.19 ± 0.14**	**2.47 ± 0.33**	**2.13 ± 0.31**	**2.26 ± 0.53**	**2.60 ± 0.58**	**2.94 ± 0.35**
*At1g28030*	*2OG-Fe(II)-oxygenase*	**2.33 ± 0.38**	1.89 ± 0.13	**2.16 ± 0.20**	1.81 ± 0.77	1.93 ± 0.24	**2.01 ± 0.31**	1.90 ± 0.13	**2.32 ± 0.37**	**2.10 ± 0.03**	**2.06 ± 0.44**	**2.10 ± 0.29**	**1.99 ± 0.51**	**2.22 ± 0.23**	**2.60 ± 0.53**
*At1g52790*	*2OG-Fe(II)-oxygenase*	**2.53 ± 0.32**	**2.00 ± 0.13**	**2.22 ± 0.30**	1.83 ± 0.29	1.82 ± 0.32	**2.21 ± 0.35**	1.00 ± 0.50	**2.14 ± 0.14**	**2.25 ± 0.18**	**2.29 ± 0.31**	**2.12 ± 0.32**	**2.46 ± 0.52**	**2.36 ± 0.17**	**2.46 ± 0.45**
*At2g44800*	*2OG-Fe(II)-oxygenase*	**2.22 ± 0.41**	**2.15 ± 0.57**	**2.14 ± 0.19**	**2.09 ± 0.42**	1.99 ± 0.24	1.93 ± 0.39	1.86 ± 0.11	**2.20 ± 0.38**	**2.15 ± 0.13**	**2.28 ± 0.42**	1.94 ± 0.43	**2.19 ± 0.60**	**2.45 ± 0.32**	**2.57 ± 0.46**
*At3g28490*	*2OG-Fe(II)-oxygenase*	**2.45 ± 0.33**	**2.18 ± 0.46**	**2.21 ± 0.19**	**2.37 ± 0.39**	1.94 ± 0.18	**2.07 ± 0.36**	1.99 ± 0.10	**2.20 ± 0.30**	**2.15 ± 0.12**	**2.26 ± 0.34**	**2.07 ± 0.29**	**2.20 ± 0.54**	**2.73 ± 0.53**	**2.62 ± 0.57**
*At1g60980*	*GA20OX4*	**2.36 ± 0.29**	**2.11 ± 0.49**	**2.13 ± 0.20**	1.94 ± 0.29	1.90 ± 0.19	**2.01 ± 0.38**	1.88 ± 0.13	**2.21 ± 0.39**	**2.13 ± 0.13**	**2.26 ± 0.42**	**2.08 ± 0.27**	**2.21 ± 0.60**	**2.86 ± 0.44**	**2.54 ± 0.44**
*At2g34555*	*GA2OX3*	**2.38 ± 0.42**	**2.18 ± 1.05**	1.50 ± 0.25	1.05 ± 0.46	0.68 ± 0.26	0.99 ± 0.48	0.84 ± 0.60	1.41 ± 0.18	**2.12 ± 0.12**	**2.17 ± 0.40**	**2.00 ± 0.29**	**2.17 ± 0.57**	**2.29 ± 0.30**	**2.11 ± 0.21**
*At1g11770*	*OGOX3*	**2.38 ± 0.31**	1.88 ± 0.13	**2.14 ± 0.20**	**2.20 ± 0.41**	1.90 ± 0.18	**2.02 ± 0.33**	1.91 ± 0.13	**2.71 ± 0.69**	**2.13 ± 0.13**	**2.44 ± 0.45**	1.98 ± 0.14	**2.19 ± 0.60**	**2.64 ± 0.50**	**2.36 ± 0.39**
*At4g20800*	*ATBBE17*	**2.36 ± 0.30**	**2.11 ± 0.47**	**2.15 ± 0.21**	**2.09 ± 0.40**	1.87 ± 0.20	**2.01 ± 0.31**	1.92 ± 0.16	**2.31 ± 0.25**	1.96 ± 0.39	**2.27 ± 0.39**	**2.05 ± 0.25**	**2.17 ± 0.58**	**2.67 ± 0.50**	**2.59 ± 0.51**
*At2g38960*	*AERO2*	**2.23 ± 0.08**	1.86 ± 0.13	**2.10 ± 0.17**	**2.07 ± 0.42**	1.85 ± 0.28	**2.18 ± 0.49**	1.71 ± 0.13	**2.33 ± 0.28**	**2.17 ± 0.12**	**2.36 ± 0.43**	**2.04 ± 0.27**	**2.19 ± 0.51**	**2.81 ± 0.44**	**2.55 ± 0.57**
*At1g32300*	*GulLO1*	**3.05 ± 0.84**	**2.15 ± 0.45**	**2.15 ± 0.19**	**2.34 ± 0.37**	1.81 ± 0.36	**2.04 ± 0.34**	1.96 ± 0.09	**2.20 ± 0.32**	**2.20 ± 0.13**	**2.25 ± 0.35**	**2.06 ± 0.28**	**2.20 ± 0.56**	**2.70 ± 0.52**	**3.01 ± 1.26**
*At5g45180*	*Monooxygenase*	**2.37 ± 0.31**	**2.19 ± 0.51**	**2.13 ± 0.15**	1.66 ± 0.85	1.73 ± 0.04	1.91 ± 0.39	1.69 ± 0.12	**2.19 ± 0.38**	**2.13 ± 0.13**	**2.25 ± 0.41**	**2.05 ± 0.28**	**2.23 ± 0.56**	**2.63 ± 0.50**	**2.74 ± 0.30**
*At1g14430*	*GOXL5*	**2.36 ± 0.29**	**2.39 ± 0.81**	**2.19 ± 0.22**	**2.24 ± 0.30**	1.87 ± 0.20	**2.04 ± 0.37**	1.90 ± 0.14	**2.32 ± 0.26**	**2.14 ± 0.11**	**2.23 ± 0.34**	**2.05 ± 0.25**	**2.34 ± 0.59**	**2.66 ± 0.50**	**2.58 ± 0.50**
*At5g09360*	*LAC14*	**3.66 ± 0.84**	1.22 ± 0.08	1.71 ± 0.36	1.66 ± 0.41	1.77 ± 0.33	1.33 ± 0.57	**2.41 ± 0.41**	**2.22 ± 0.28**	**2.07 ± 0.18**	1.95 ± 0.01	1.94 ± 0.06	**3.35 ± 0.88**	**2.41 ± 0.21**	**1.56 ± 0.77**
*At3g45810*	*RBOHJ*	**2.00 ± 0.37**	1.87 ± 0.15	**2.15 ± 0.21**	**2.25 ± 0.31**	1.83 ± 0.20	1.97 ± 0.31	1.93 ± 0.13	**2.15 ± 0.33**	**2.12 ± 0.13**	**2.17 ± 0.29**	**2.02 ± 0.27**	**2.14 ± 0.55**	**2.80 ± 0.51**	**2.36 ± 0.47**
*At1g65990*	*PRXIIA*	**2.21 ± 0.04**	**2.40 ± 0.81**	**2.20 ± 0.21**	**2.10 ± 0.53**	1.85 ± 0.20	1.81 ± 0.16	1.83 ± 0.12	**2.02 ± 0.21**	**2.13 ± 0.13**	**2.25 ± 0.35**	**2.03 ± 0.28**	**2.17 ± 0.58**	**2.62 ± 0.50**	**2.64 ± 0.10**
*At1g24110*	*PER6*	**2.46 ± 0.39**	**2.14 ± 0.57**	**2.19 ± 0.21**	**2.26 ± 0.34**	1.76 ± 0.42	1.86 ± 0.37	1.84 ± 0.26	**2.41 ± 0.36**	**2.04 ± 0.27**	**2.20 ± 0.48**	1.50 ± 0.28	**2.29 ± 0.78**	**2.23 ± 0.46**	**2.36 ± 0.38**
*At3g42570*	*Peroxidase*	**2.35 ± 0.31**	1.93 ± 0.17	**2.09 ± 0.22**	1.40 ± 0.75	1.89 ± 0.22	**1.99 ± 0.30**	1.87 ± 0.05	**2.36 ± 0.28**	**2.10 ± 0.11**	**2.28 ± 0.42**	**2.07 ± 0.27**	**2.22 ± 0.59**	**2.48 ± 0.37**	**2.21 ± 0.42**

**Table 3 biomolecules-12-01824-t003:** Group C. Genes characterized by a strong upregulation in shoots at 24 h. Genes were selected based on a fold change ≥2 or ≤0.5 in at least one time point. Values are expressed as the mean and standard deviation (±), values with a fold change ≥2 or ≤0.5 are indicated as boldfaced numbers.

		ROOT							SHOOT						
Gene Locus	Gene Code	10 min	1 h	2 h	4 h	24 h	48 h	96 h	10 min	1 h	2 h	4 h	24 h	48 h	96 h
*At1g26390*	*ATBBE4*	**2.06 ± 0.56**	1.80 ± 1.49	1.21 ± 0.99	0.58 ± 0.27	0.77 ± 0.09	1.49 ± 0.69	0.83 ± 0.45	**3.22 ± 1.39**	**2.40 ± 0.55**	1.22 ± 0.56	1.84 ± 1.19	**6.89 ± 4.90**	1.26 ± 0.71	0.52 ± 0.64
*At1g26410*	*ATBBE6*	**0.29 ± 0.10**	1.46 ± 1.02	0.87 ± 0.52	0.81 ± 0.18	0.60 ± 0.05	1.44 ± 0.18	1.22 ± 1.22	0.59 ± 0.12	1.72 ± 0.15	0.73 ± 0.50	0.60 ± 0.31	**2.85 ± 3.47**	0.55 ± 0.13	**0.32 ± 0.26**
*At4g21830*	*MSRB7*	0.97 ± 0.15	1.03 ± 0.17	0.92 ± 0.16	0.88 ± 0.07	0.90 ± 0.24	0.87 ± 0.43	0.84 ± 0.14	1.88 ± 1.83	0.67 ± 0.49	0.81 ± 0.07	1.75 ± 0.35	**2.36 ± 1.14**	1.27 ± 1.08	0.85 ± 0.21
*At5g05340*	*PRX52*	0.87 ± 0.09	1.11 ± 0.24	1.21 ± 0.30	0.88 ± 0.26	1.84 ± 1.08	0.91 ± 0.18	1.08 ± 0.25	1.40 ± 0.18	0.95 ± 0.05	0.92 ± 0.18	0.75 ± 0.11	**2.36 ± 2.79**	1.70 ± 0.28	0.91 ± 0.49

**Table 4 biomolecules-12-01824-t004:** Group D. Genes characterized by a strong upregulation in shoots at 48 h. Genes were selected based on a fold change ≥2 or ≤0.5 in at least one time point. Values are expressed as the mean and standard deviation (±), values with a fold change ≥2 or ≤0.5 are indicated as boldfaced numbers.

		ROOT							SHOOT						
Gene Locus	Gene Code	10 min	1 h	2 h	4 h	24 h	48 h	96 h	10 min	1 h	2 h	4 h	24 h	48 h	96 h
*At2g38240*	*AO4*	0.90 ± 0.12	0.99 ± 0.11	0.79 ± 0.28	1.00 ± 0.09	1.83 ± 0.79	0.87 ± 0.17	1.17 ± 0.20	0.61 ± 0.12	1.91 ± 0.89	0.64 ± 0.19	0.98 ± 0.70	0.38 ± 0.25	**2.56 ± 2.12**	1.08 ± 0.05
*At5g56470*	*GULLO7*	**2.00 ± 0.54**	1.28 ± 0.69	**2.40 ± 1.04**	1.34 ± 0.42	1.96 ± 0.18	1.46 ± 0.92	1.03 ± 0.24	1.48 ± 0.34	1.63 ± 0.57	**2.21 ± 0.46**	**2.06 ± 0.47**	**2.12 ± 0.68**	**2.71 ± 0.59**	**2.12 ± 0.55**
*At5g23990*	*FRO5*	1.44 ± 0.31	0.60 ± 0.46	0.85 ± 0.03	0.81 ± 0.27	1.44 ± 0.58	**2.03 ± 0.35**	1.98 ± 0.09	**2.17 ± 0.31**	**2.13 ± 0.11**	**2.22 ± 0.38**	1.91 ± 0.03	**2.10 ± 0.63**	**2.70 ± 0.53**	1.95 ± 0.24
*At3g56350*	*MSD2*	0.60 ± 0.07	0.60 ± 0.32	0.71 ± 0.14	**2.13 ± 0.51**	1.41 ± 0.03	0.63 ± 0.22	1.23 ± 0.16	0.56 ± 0.12	1.64 ± 0.14	0.93 ± 0.27	0.91 ± 0.16	1.10 ± 0.85	**2.89 ± 2.88**	1.78 ± 0.16
*At3g50990*	*PER36*	1.09 ± 0.08	1.90 ± 0.15	**2.29 ± 0.36**	**2.53 ± 0.28**	**2.23 ± 0.50**	**2.25 ± 1.21**	1.36 ± 0.36	**3.76 ± 1.48**	**2.21 ± 0.20**	**2.77 ± 0.30**	**2.08 ± 0.27**	0.98 ± 0.12	**5.27 ± 3.73**	**2.11 ± 0.10**

**Table 5 biomolecules-12-01824-t005:** Group E. Genes downregulated in shoots at 24 h. Genes selected based on a fold change ≥2 or ≤0.5 in at least one time point. Values are expressed as the mean and standard deviation (±), values with a fold change ≥2 or ≤0.5 are indicated as boldfaced numbers.

		ROOT							SHOOT						
Gene Locus	Gene Code	10 min	1 h	2 h	4 h	24 h	48 h	96 h	10 min	1 h	2 h	4 h	24 h	48 h	96 h
*At1g18490*	*PCO3*	0.89 ± 0.24	1.38 ± 0.33	1.62 ± 0.35	0.92 ± 0.18	1.32 ± 0.24	0.78 ± 0.07	1.01 ± 0.12	0.74 ± 0.12	0.75 ± 0.17	1.15 ± 0.58	1.27 ± 0.57	**0.49 ± 0.10**	1.01 ± 0.48	1.18 ± 0.25

**Table 6 biomolecules-12-01824-t006:** Validation of gene microarrays by qPCR. Fold change of NNMF expression vs. GMF expression of *Arabidopsis thaliana* exposed for 96 h. Values are expressed as means of at least three biological replicates (±standard deviation).

Gene Locus	FC Roots	FC Shoots
*At2g18140*	0.10 ± 0.02	5.95 ± 0.12
*At1g32300*	2.60 ± 0.43	3.08 ± 0.25
*At3g56350*	0.70 ± 0.10	2.84 ± 0.34

**Table 7 biomolecules-12-01824-t007:** Time-course analysis of H_2_O_2_ production in *Arabidopsis thaliana* under GMF and NNMF. Values are expressed as mmol H_2_O_2_ and fold change (NNMF/GMF). Within the same column, the letters indicate statistical difference (*p* < 0.05) among shoots (lowercase letters) or roots (uppercase letters), as measured by one-way ANOVA followed by Tukey’s test. *p* values refer to the statistical difference in H_2_O_2_ content of samples grown under GMF and NMMF with the same sampling timing. For further statistical information see Appendix A.

Organ	Timing	GMFmmol H_2_O_2_	NNMFmmol H_2_O_2_	NNMF/GMF	*p* Value
Shoots	24 h	0.092 ± 0.006 ^a^	0.025 ± 0.003 ^a^	0.276 ± 0.067	<0.001
	48 h	0.047 ± 0.005 ^b^	0.025 ± 0.006 ^a^	0.528 ± 0.002	0.001
	96 h	0.016 ± 0.001 ^c^	0.013 ± 0.003 ^b^	0.805 ± 0.006	0.006
Roots	24 h	0.051 ± 0.006 ^A^	0.014 ± 0.001 ^A^	0.274 ± 0.034	<0.001
	48 h	0.040 ± 0.007 ^A^	0.025 ± 0.001 ^B^	0.625 ± 0.031	0.011
	96 h	0.050 ± 0.005 ^A^	0.041 ± 0.003 ^C^	0.834 ± 0.080	0.013

**Table 8 biomolecules-12-01824-t008:** Time-course analysis of total polyphenol contents in *Arabidopsis thaliana* under GMF and NNMF. Values are expressed as µg g^−1^ FW or fold change (NNMF/GMF). Within the same column, letters indicate statistical difference (*p* < 0.05) among shoots (lowercase letters) or roots (uppercase letters), as measured by one-way ANOVA followed by Tukey’s test. *p* value refers to the statistical difference in polyphenol content between plants grown under GMF or NMMF with the same sampling timing. For further statistical information see Appendix A.

**Organ**	**Timing**	**GMF** **µg g^−1^ FW**	**NNMF** **µg g^−1^ FW**	**NNMF/GMF**	***p* Value**
Shoots	24 h	260.753 (6.005) ^b^	707.880 (20.365) ^b^	2.714 (0.076)	<0.001
	48 h	557.101 (20.729) ^a^	1003.704 (22.784) ^a^	1.801 (0.069)	<0.001
	96 h	204.548 (3.458) ^c^	125.134 (3.532) ^c^	0.611 (0.018)	<0.001
Roots	24 h	109.149 (9.445) ^AB^	210.005 (3.922) ^B^	1.924 (0.025)	<0.001
	48 h	85.917 (6.197) ^B^	264.412 (7.945) ^A^	3.077 (0.061)	<0.001
	96 h	122.059 (9.173) ^A^	87.315 (1.791) ^C^	0.715 (0.024)	<0.001

**Table 9 biomolecules-12-01824-t009:** Fold change values of NNMF vs. GMF polyphenols in *Arabidopsis thaliana* time-course experiments from Appendix A. Fold change values are expressed along with standard deviation and calculated from the quantitative data of Appendix A.

#	Compound(s)	Roots			Shoots		
		24 h	48 h	96 h	24 h	48 h	96 h
**1**	Daidzein-galactoside	1.392 (0.026)	2.23 (0.083)	1.542 (0.032)	8.542 (0.149)	9.238 (0.195)	0.403 (0.007)
**2**	Genistein-galactoside				0.656 (0.02)	0.325 (0.009)	0.785 (0.011)
**3**	Dihydrogenistein-galactoside				11.568 (0.164)	0.127 (0.003)	0.874 (0.026)
**4**	Genistein-glucoside				1.424 (0.032)	0.746 (0.013)	1.587 (0.041)
**5**	Dihydrogenistein-glucoside				3.611 (0.056)	1.443 (0.051)	0.885 (0.025)
**6**	Dihydrogenistein-glucuronide	3.251 (0.125)	4.071 (0.065)	5.032 (0.127)	0.712 (0.013)	6.491 (0.108)	0.715 (0.027)
**7**	Dihydrokaempferol-glucoside				1.371 (0.053)	1.507 (0.056)	0.845 (0.019)
**8**	Daidzein-glucoside	1.353 (0.03)	2.986 (0.063)	0.624 (0.024)	2.134 (0.034)	3.024 (0.1)	0.664 (0.008)
**9**	Dihydromyricetin-diglucoside	2.214 (0.042)	4.495 (0.125)	0.475 (0.013)	3.111 (0.045)	1.452 (0.03)	0.614 (0.016)
**10**	Quercetin-rutinoside	2.232 (0.088)	1.087 (0.027)	0.368 (0.004)	1.219 (0.046)	1.902 (0.043)	0.272 (0.006)
**11**	Dihydroquercetin-rutinoside	1.883 (0.042)	0.316 (0.008)	4.69 (0.145)	0.763 (0.025)	9.222 (0.176)	0.236 (0.007)
**12**	Dihydrohesperetin-neohesperidoside				5.245 (0.199)	1.489 (0.048)	0.27 (0.007)
**13**	Kaempferol-rutinoside	6.88 (0.114)	2.932 (0.108)	0.996 (0.025)		0.763 (0.015)	1.054 (0.033)
**14**	Quercetin-sambubioside	2.126 (0.075)	4.211 (0.08)	0.826 (0.03)	4.615 (0.092)	2.284 (0.04)	0.489 (0.018)
**15**	Quercetin-dirhamnopyranoside	7.846 (0.192)	2.903 (0.101)	0.973 (0.031)	0.933 (0.017)	7.443 (0.129)	0.692 (0.007)
**16**	Dihydroquercetin-sambubioside	3.19 (0.099)	5.405 (0.132)	0.338 (0.004)	3.303 (0.036)	3.638 (0.104)	0.407 (0.012)
**17**	Isorhamnetin-rutinoside	1.538 (0.05)	3.06 (0.1)	0.533 (0.016)	2.299 (0.074)	2.38 (0.081)	0.404 (0.014)
**18**	Myricetin-rutinoside	1.23 (0.022)	3.246 (0.111)	1.072 (0.035)	4.066 (0.09)	2.209 (0.076)	0.374 (0.007)
**19**	Dihydroquercetin-galactoside	1.178 (0.042)	3.57 (0.103)	0.597 (0.016)	3.811 (0.072)	1.367 (0.034)	1.171 (0.038)
**20**	Kaempferol-sambubioside				3.914 (0.11)	2.425 (0.073)	0.651 (0.019)
**21**	Dihydromyricetin-sambubioside				1.423 (0.017)	6.134 (0.215)	0.671 (0.015)
**22**	Myricetin-diglucoside				1.206 (0.02)	2.128 (0.071)	3.415 (0.049)
**23**	Isorhamnetin-sambubioside	4.069 (0.138)	5.173 (0.067)	1.371 (0.019)	3.297 (0.099)	2.368 (0.076)	0.571 (0.014)
**24**	Quercetin-rhamnoside	1.699 (0.019)	5.881 (0.157)	0.121 (0.004)	1.088 (0.023)	2.791 (0.062)	1.139 (0.018)
**25**	Daidzein-glucuronide	1.889 (0.069)	4.057 (0.113)	0.62 (0.024)	0.373 (0.008)	0.35 (0.01)	0.741 (0.029)
**26**	Dihydroquercetin-glucoside	2.536 (0.078)	5.348 (0.129)	1.658 (0.052)	5.106 (0.178)	0.168 (0.006)	1.199 (0.016)
**27**	Unknown	1.862 (0.068)	2.412 (0.072)	0.257 (0.006)	0.27 (0.005)	0.264 (0.004)	5.933 (0.105)
**28**	Dihydromyricetin-galactoside	1.952 (0.046)	2.301 (0.062)	0.248 (0.007)	0.626 (0.017)	0.206 (0.007)	0.574 (0.016)
**29**	Quercetin-galactoside	1.933 (0.021)	6.731 (0.21)	0.578 (0.008)	3.349 (0.044)	0.41 (0.004)	0.895 (0.017)
**30**	Isorhamnetin-galactoside	8.576 (0.097)	4.028 (0.048)	0.545 (0.016)			
**31**	Dihydromyricetin-rutinoside				1.871 (0.055)	1.216 (0.031)	1.017 (0.015)
**32**	Isorhamnetin-glucoside	0.621 (0.01)	3.002 (0.077)	0.638 (0.011)			
**33**	Quercetin-glucoside	0.582 (0.013)	9.052 (0.114)	0.443 (0.009)	0.224 (0.004)	0.737 (0.02)	0.562 (0.02)
**34**	Dihydroquercetin-diglucoside				0.989 (0.011)	2.824 (0.102)	0.703 (0.017)
**35**	Dihydromyricetin-glucoside	1.74 (0.038)	1.481 (0.017)	2.263 (0.079)	3.228 (0.082)	1.366 (0.036)	1.8 (0.031)
**36**	Dihydromyricetin-rhamnoside	1.703 (0.052)	3.06 (0.082)	0.917 (0.016)	4.886 (0.103)	1.459 (0.039)	1.984 (0.041)
**37**	Genistein	1.418 (0.027)	2.792 (0.083)	1.166 (0.033)	4.146 (0.148)	1.532 (0.028)	1.176 (0.019)
**38**	Daidzein	8.564 (0.294)	2.658 (0.083)	0.795 (0.021)	2.074 (0.036)	3.164 (0.119)	0.481 (0.015)
**39**	Kaempferol				4.634 (0.121)	5.079 (0.089)	0.213 (0.004)
**40**	Dihydrogenistein	4.147 (0.153)	6.492 (0.105)	0.485 (0.018)	4.153 (0.085)	1.625 (0.046)	0.452 (0.011)
**41**	Dihydrokaempferol				7.355 (0.087)	13.806 (0.438)	5.086 (0.106)
**42**	Quercetin	1.26 (0.031)	1.316 (0.014)	0.524 (0.011)	4.703 (0.163)	9.87 (0.229)	1.44 (0.022)
**43**	Myricetin	1.658 (0.021)	7.513 (0.157)	0.831 (0.018)	4.685 (0.147)	1.552 (0.019)	0.494 (0.016)
**44**	Isorhamnetin	1.726 (0.026)	0.022 (0)	0.799 (0.022)	4.673 (0.062)	1.612 (0.027)	0.452 (0.004)
**45**	Dihydroquercetin				3.936 (0.147)	2.938 (0.107)	0.616 (0.013)

## Data Availability

Data are available as Appendix A and further data can be provided upon request.

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
