# Peer review of "Transcriptomics and Metabolomics of Reactive Oxygen Species Modulation in Near-Null Magnetic Field-Induced Arabidopsis thaliana"

_biomolecules, 2022, doi:10.3390/biom12121824_

Round 1
Reviewer 1 Report
In the manuscript titled “Genomics and Metabolomics of Reactive Oxygen Species Modulation in Near Null Magnetic Field-Induced Arabidopsis thaliana”, the authors found that GMF modulated the redox balance by regulating the antioxidant enzymes and polyphenols in Arabidopsis. Generally the manuscript is well writen, and the experiments were well-designed. Below I have some suggestions and I listed below:
1. I suggested that the KEGG pathway-based enrichment analyses should be performed, and the authors can screen whether the flavonoid biosynthesis and other pathways are enriched or not.
2. I suggested that H2O2 content should be detected in roots and shoots by treatments for 10 min, 1 h, 2 h, and 4 h, and the correlation coefficients between H2O2 content and ROS-related gene expressions can be analyzed. Then, the important genes will be studied in future.
3. In Figure 2, the authors can clearly separate the genes-related ROS generation and scavenging.
4. In Figure 4, the authors can show the polyphenol metabolism, and mark the changed genes and compounds in the pathway.
5. Pay attention to the writing style, such as line 14 “10min” “96h”, in the whole article.
Author Response
We thank the reviewer for the critical reading of our manuscript. We addressed the questions. We have amended the text and figures accordingly each time it was feasible for us. We have used track changes in the revised manuscript to highlight all the changes we have made. A point-by-point response to the reviewer’ comments is provided below. We believe these changes will definitely help improve the manuscript, and we hope that our arguments will convince the reviewers and editors that it is now acceptable for publication.
REVIEWER 1
In the manuscript titled “Genomics and Metabolomics of Reactive Oxygen Species Modulation in Near Null Magnetic Field-Induced Arabidopsis thaliana”, the authors found that GMF modulated the redox balance by regulating the antioxidant enzymes and polyphenols in Arabidopsis. Generally the manuscript is well written, and the experiments were well-designed.
R: we thank the reviewer for the appreciation of our work
Below I have some suggestions and I listed below:
- I suggested that the KEGG pathway-based enrichment analyses should be performed, and the authors can screen whether the flavonoid biosynthesis and other pathways are enriched or not.
R: We thank the reviewer for this suggestion. Based on the comments of reviewer 2 we have moved the Supplementary Figure S1 to the main text in order to provide a better picture of the gene involvement. Running a KEGG pathway-based enrichment analysis would require to consider the different timings and the different organs. This would require a major revision of the manuscript which we understood it was not your choice of revision. Therefore we would prefer not to run this analysis. However, we will keep your interesting suggestion for further studies.
- I suggested that H2O2 content should be detected in roots and shoots by treatments for 10 min, 1 h, 2 h, and 4 h, and the correlation coefficients between H2O2 content and ROS-related gene expressions can be analyzed. Then, the important genes will be studied in future.
R: we thank the reviewer for this comments. We tried to run H2O2 analysis from 10 min to 2 h but the material was not enough for the sensitivity of the reaction test that we used; the standard error was always higher than the mean itself. Therefore, we decided to keep only the timings that allowed us to evaluate the H2O2 content with a sound and significant statistical evaluation of data. We are currently developing methods for fixing samples for confocal studies and we will report these results in the next publication.
A sentence explains the reason why 10 min to 8 h sampling was not done.
- In Figure 2, the authors can clearly separate the genes-related ROS generation and scavenging.
R: the reviewer is right and in the revised version we outlined the ROS generating genes in boldface in other to distinguish them from the ROS scavenging. The figure caption has been updated as well.
- In Figure 4, the authors can show the polyphenol metabolism, and mark the changed genes and compounds in the pathway.
R: we thank the reviewer for this suggestion. Figure 4 has been completely redrawn in order to show the relationship between the antioxidant polyphenols and the associated genes. The text has been updated in order to guide the reader in the interpretation of data.
- Pay attention to the writing style, such as line 14 “10min” “96h”, in the whole article.
R: the writing style has been edited
Reviewer 2 Report
Dear Authors,
I have reviewed your manuscript "Genomics and Metabolomics of Reactive Oxygen Species Modulation in Near Null Magnetic Field-Induced Arabidopsis thaliana", submitted for publication in Biomolecules.
I find that your manuscript is in a very good shape. Your research design is appropriate, the results are interesting and properly discussed, and the quality of English language is very good. However, I have some comments which I hope can help you make your manuscript even better. The suggestions for revision are as follows:
· Genomics vs. Transcriptomics - The term "genomics" refers to the structure of a genome regardless the transcriptional activity of particular genes, and therefore the research that you performed here, on how NNMF affects transcription of particular genes, is a transcriptomic, not a genomic research. This should be more accurately stated in the title of your manuscript (literally its first word), and also at multiple points throughout the manuscript (e.g., line 23, line 569, etc.).
· Arabidopsis - the word Arabidopsis is written in plain text throughout the manuscript. Please put Arabidopsis in italic letters systematically everywhere throughout the manuscript text.
· Introduction:
o lines 47-49: "Thus, the primary effect of a MF..." - this sentence is a completely unnecessary repetition of a previous statement, I recommend to delete it.
o line 58: "Therefore, the aim of this work..." - this should be put in a new paragraph. The last paragraph of the Introduction should always be narrowly related to the aim of the present work and should be physically separated from the previous text, especially when the previous text (lines 54-58) is not very much related to the current work.
· Material & Methods:
o line 91: How old were the seedlings at the time of their exposure to NNMF?
o Section 2.2, as well: Did you measure the actual strength of the magnetic field within the triaxial Helmholtz coils at the place where the plants were kept in this experiment, or do you just refer to it as near-null based on previous research? Either way it should be stated explicitly within the section 2.2.
o volume units - mililiter and microliter are written randomly with a lowercase or uppercase L throughout the text (e.g., ml and mL, or µl and µL). Please opt for one of them and stick to it consistently (I recommend the uppercase L).
o reference 27 is the same as reference 21. Please revise.
o line 185-186: the reference for the Benjamini-Hochberg multiple testing correction would be welcome.
· Results:
o Supplementary Figure 1 - I find that the results of the Gene Ontology analysis can be considered important enough to be included in the manuscript as a regular figure instead of a supplementary figure, especially since there are already too many supplementary files. This is not obligatory, but the Authors might reconsider that option and decide whether they find it reasonable or not.
o lines 217 and 255: "genes that are not regulated or downregulated in roots and upregulated in shoots". Please rephrase, both in the text of the Results, and in the caption to Table 1. What is meant by "not regulated"? That they are neither upregulated nor downregulated in the roots, or that they are not expressed in the roots at all? Either way the term "not regulated" is inappropriate.
o the differentially expressed genes (DEGs) - I have several remarks regarding these:
§ the term "differentially expressed genes", abbreviated as "DEGs", is often used in transcriptomic research. Using it might make your text more elegant at a couple of points. This is just a casual remark in case you find it useful, there is no big need to revise the text.
§ the heatmap in Figure 1 shows that in your research, you discovered a lot more DEGs than the ones given in the Tables, especially in the Tables 3-5. Why is that? Are only the DEGs with known functions displayed in the Tables? The explanation for this should be given within the captions for each Table.
§ Another question concerns the functions of the identified DEGs. All the DEGs that are mentioned within the text of the Results, actually encode enzymes. Is it possible that all the DEGs in your research encode enzymes, and no genes encoding, for instance, transcription factors, were differentially expressed? Or did the Authors filter out the DEGs with functions other than enzymes? Please provide brief commentary/explanation within the manuscript text.
o line 320 and 373: it is Tukey's test, not "Tuckey's test"
o line 365 - please add the word "total" ("... content of the total identified polyphenols...")
o line 367 - there is a typo ("GMG" instead of "GMF")
o Table 8 - I have a question regarding these results: How do you explain the time-course variations in total polyphenol content in control plants (subjected to GMF)? These plants are literally just growing there in normal conditions, as all plants do everywhere, and you measured their total polyphenol content at different time points since the beginning of application of NNMF to the other group of plants. If you do not have a proper explanation for this puzzling result, you might consider omitting it from the manuscript.
o Table 9 - Did you consider displaying these data in a heatmap, instead of a table? A heatmap might be more readily informative on an intuitive level, although a table contains technically more information. Of course, the decision about this should be up to the Authors.
· Discussion and Conclusions:
o Although your results are quite well discussed, I was hoping to read a more solid explanation as to why most antioxidant mechanisms seem to be more active in the NNMF plants than in control plants, even though the endogenous levels of ROS are lower than in GMF? Since GMF is a normal (natural) condition, would it still be appropriate to call it a stressing condition compared to the (unnatural) NNMF? Or, alternatively, is it possible that the plant homeostatic mechanisms "perceive" NNMF and interpret it as a stress-like condition (even though NNMF, per sé, brings little or no actual stress to plants), and then react to it by activating ROS scavenging and other antioxidant mechanisms which then end up decreasing the levels of ROS in plant tissue even below the normal (control) levels? Commentary on that particular issue within your Discussion section would make your Discussion more informative and it would provide a more comprehensive explanation for the phenomena that you observed in your work.
o The inclusion of such commentary in your Discussion would provide a basis for a more solid and focused Conclusions section.
o Other minor remarks regarding the Discussion section:
§ line 482: gene transcription is always DNA-templated. You actually wanted to say that this gene encodes a transcription factor (which you actually stated immediately afterwards, in line 485). Kindly revise.
§ lines 494-495: "oligosaccharide oxidase that oxidizes", not "oxidases that oxidize". If you carefully read the sentence, it should be in singular, not in plural.

Author Response
We thank the reviewer for the critical reading of our manuscript. We addressed the questions. We have amended the text and figures accordingly each time it was feasible for us. We have used track changes in the revised manuscript to highlight all the changes we have made. A point-by-point response to the reviewer’ comments is provided below. We believe these changes will definitely help improve the manuscript, and we hope that our arguments will convince the reviewers and editors that it is now acceptable for publication.
REVIEWER 2
I have reviewed your manuscript "Genomics and Metabolomics of Reactive Oxygen Species Modulation in Near Null Magnetic Field-Induced Arabidopsis thaliana", submitted for publication in Biomolecules. I find that your manuscript is in a very good shape. Your research design is appropriate, the results are interesting and properly discussed, and the quality of English language is very good.
R: we are grateful to the reviewer for the appreciation of our work
However, I have some comments which I hope can help you make your manuscript even better. The suggestions for revision are as follows:
- Genomics vs. Transcriptomics - The term "genomics" refers to the structure of a genome regardless the transcriptional activity of particular genes, and therefore the research that you performed here, on how NNMF affects transcription of particular genes, is a transcriptomic, not a genomic research. This should be more accurately stated in the title of your manuscript (literally its first word), and also at multiple points throughout the manuscript (e.g., line 23, line 569, etc.).
R: we thank the reviewer for this important remark. The term genomics has been replaced with transcriptomics.
- Arabidopsis - the word Arabidopsis is written in plain text throughout the manuscript. Please put Arabidopsis in italic letters systematically everywhere throughout the manuscript text.
R: the term Arabidopsis in normal text has been replaced with A. thaliana in italics throughout all text.
- Introduction:
- lines 47-49: "Thus, the primary effect of a MF..." - this sentence is a completely unnecessary repetition of a previous statement, I recommend to delete it.
R: the sentence has been deleted
o line 58: "Therefore, the aim of this work..." - this should be put in a new paragraph. The last paragraph of the Introduction should always be narrowly related to the aim of the present work and should be physically separated from the previous text, especially when the previous text (lines 54-58) is not very much related to the current work.
R: a new paragraph has been created as suggested
- Material & Methods:
o line 91: How old were the seedlings at the time of their exposure to NNMF?
R: the age of plants is described in 2.1 and we added “seeded 134 h before” to remind the reader the age of plants as suggested by the reviewer.
o Section 2.2, as well: Did you measure the actual strength of the magnetic field within the triaxial Helmholtz coils at the place where the plants were kept in this experiment, or do you just refer to it as near-null based on previous research? Either way it should be stated explicitly within the section 2.2.
R: the MF intensity is constantly measured as stated in 2.2 “Real-time monitoring of the MF in the plant exposure chamber was achieved with a three-axis magnetic field sensor (model Mag-03, Bartington Instruments, Oxford, U.K.) that was placed at the geometric center of the Helmholtz coils”
- volume units - mililiter and microliter are written randomly with a lowercase or uppercase L throughout the text (e.g., ml and mL, or µl and µL). Please opt for one of them and stick to it consistently (I recommend the uppercase L).
R: we thank the reviewer for noticing this. The unit L has been replaced throughout the text.
o reference 27 is the same as reference 21. Please revise.
R: we thank the reviewer for noticing this. The reference has been modified.
o line 185-186: the reference for the Benjamini-Hochberg multiple testing correction would be welcome.
R: the reference has been inserted
- Results:
o Supplementary Figure 1 - I find that the results of the Gene Ontology analysis can be considered important enough to be included in the manuscript as a regular figure instead of a supplementary figure, especially since there are already too many supplementary files. This is not obligatory, but the Authors might reconsider that option and decide whether they find it reasonable or not.
R: we thank the reviewer for this comment. The Supplementary Figure S1 has been inserted in the text as the new Figure 1 and a new caption has been added. Therefore, all subsequent numbering of figures has been updated.
o lines 217 and 255: "genes that are not regulated or downregulated in roots and upregulated in shoots". Please rephrase, both in the text of the Results, and in the caption to Table 1. What is meant by "not regulated"? That they are neither upregulated nor downregulated in the roots, or that they are not expressed in the roots at all? Either way the term "not regulated" is inappropriate.
R: thank you for this remark. In the revised text we defined “not regulated” as genes whose expression is below 2 or higher than 0.5 Fold change.
o the differentially expressed genes (DEGs) - I have several remarks regarding these:
- the term "differentially expressed genes", abbreviated as "DEGs", is often used in transcriptomic research. Using it might make your text more elegant at a couple of points. This is just a casual remark in case you find it useful, there is no big need to revise the text.
R: thank you for this useful remark. We changed to DEG(s) when possible
- the heatmap in Figure 1 shows that in your research, you discovered a lot more DEGs than the ones given in the Tables, especially in the Tables 3-5. Why is that? Are only the DEGs with known functions displayed in the Tables? The explanation for this should be given within the captions for each Table.
R: the heatmap represent all expressed genes, whereas the tables, as indicated in the heading, only report those genes where at least in one time point the FC is > 2 or < 0.5.
- Another question concerns the functions of the identified DEGs. All the DEGs that are mentioned within the text of the Results, actually encode enzymes. Is it possible that all the DEGs in your research encode enzymes, and no genes encoding, for instance, transcription factors, were differentially expressed? Or did the Authors filter out the DEGs with functions other than enzymes? Please provide brief commentary/explanation within the manuscript text.
R: Thanks for this remark. We specified in the text that genes were filtered considering those coding for enzymes involved in oxidative stress.
o line 320 and 373: it is Tukey's test, not "Tuckey's test"
R: the term Tukey has been edited
o line 365 - please add the word "total" ("... content of the total identified polyphenols...")
R: the sentence has been corrected
o line 367 - there is a typo ("GMG" instead of "GMF")
R: thank you for noticing this typo, which has been corrected
o Table 8 - I have a question regarding these results: How do you explain the time-course variations in total polyphenol content in control plants (subjected to GMF)? These plants are literally just growing there in normal conditions, as all plants do everywhere, and you measured their total polyphenol content at different time points since the beginning of application of NNMF to the other group of plants. If you do not have a proper explanation for this puzzling result, you might consider omitting it from the manuscript.
R: during plant development both gene expression and the content of phenolic compounds changes. This is due to the increased photosynthetic activity and the differentiation of tissues. In this way, it is quite normal to observe developmental metabolic variations. This is why we compared the NNMF plants with the GMF plants. New language addresses to this point
o Table 9 - Did you consider displaying these data in a heatmap, instead of a table? A heatmap might be more readily informative on an intuitive level, although a table contains technically more information. Of course, the decision about this should be up to the Authors.
R: actually the heatmap of figure 5A is the heatmap representation of Table 9
- Discussion and Conclusions:
o Although your results are quite well discussed, I was hoping to read a more solid explanation as to why most antioxidant mechanisms seem to be more active in the NNMF plants than in control plants, even though the endogenous levels of ROS are lower than in GMF? Since GMF is a normal (natural) condition, would it still be appropriate to call it a stressing condition compared to the (unnatural) NNMF? Or, alternatively, is it possible that the plant homeostatic mechanisms "perceive" NNMF and interpret it as a stress-like condition (even though NNMF, per sé, brings little or no actual stress to plants), and then react to it by activating ROS scavenging and other antioxidant mechanisms which then end up decreasing the levels of ROS in plant tissue even below the normal (control) levels? Commentary on that particular issue within your Discussion section would make your Discussion more informative and it would provide a more comprehensive explanation for the phenomena that you observed in your work.
R: we thank the reviewer for this constructive observation. Actually, the GMF is not constant and plants in different parts of the world experience different inclination and intensity of the GMF. Even during the same day the values are not constant. This is why we consider any variation of MF a stress factor, as it is for temperature or light. Our NNMF system is the “contro” situation of the GMF and helped us to evaluate what happens in “normal” conditions. We added a sentence in the conclusions to better stress this point.
o The inclusion of such commentary in your Discussion would provide a basis for a more solid and focused Conclusions section.
R: we agree (see above)
o Other minor remarks regarding the Discussion section:
- line 482: gene transcription is always DNA-templated. You actually wanted to say that this gene encodes a transcription factor (which you actually stated immediately afterwards, in line 485). Kindly revise.
R: the sentence has been deleted
- lines 494-495: "oligosaccharide oxidase that oxidizes", not "oxidases that oxidize". If you carefully read the sentence, it should be in singular, not in plural.
R: the sentence has been revised as suggested